# Lipid bodies containing oxidatively truncated lipids block antigen cross-presentation by dendritic cells in cancer

Filippo Veglia[1], Vladimir A. Tyurin[2], Dariush Mohammadyani[2,3], Maria Blasi[4], Elizabeth K. Duperret[1,5], Laxminarasimha Donthireddy[1], Ayumi Hashimoto[1], Alexandr Kapralov[2], Andrew Amoscato[2], Roberto Angelini[2], Sima Patel[1], Kevin Alicea-Torres[1], David Weiner[1,5], Maureen E. Murphy[6], Judith Klein-Seetharaman [2], Esteban Celis[7], Valerian E. Kagan[2] & Dmitry I. Gabrilovich[1]

Cross-presentation is a critical function of dendritic cells (DCs) required for induction of antitumor immune responses and success of cancer immunotherapy. It is established that tumor-associated DCs are defective in their ability to cross-present antigens. However, the mechanisms driving these defects are still unknown. We find that impaired cross-presentation in DCs is largely associated with defect in trafficking of peptide–MHC class I (pMHC) complexes to the cell surface. DCs in tumor-bearing hosts accumulate lipid bodies (LB) containing electrophilic oxidatively truncated (ox-tr) lipids. These ox-tr-LB, but not LB present in control DCs, covalently bind to chaperone heat shock protein 70. This interaction prevents the translocation of pMHC to cell surface by causing the accumulation of pMHC inside late endosomes/lysosomes. As a result, tumor-associated DCs are no longer able to stimulate adequate CD8 T cells responses. In conclusion, this study demonstrates a mechanism regulating cross-presentation in cancer and suggests potential therapeutic avenues.

[1] Translational Tumor Immunology Program, The Wistar Institute, Philadelphia, PA 19104, USA. [2] Department of Environmental and Occupational Health, University of Pittsburgh, Pittsburgh, PA 15219, USA. [3] Thomas C. Jenkins Department of Biophysics, Johns Hopkins University, Baltimore, MD 21218, USA. [4] Duke University Medical Center, Durham, NC 27710, USA. [5] Vaccine Center, The Wistar Institute, Philadelphia, PA 19104, USA. [6] Program in Molecular and Cellular Oncogenesis, The Wistar Institute, Philadelphia, PA 19104, USA. [7] Cancer Immunology, Inflammation and Tolerance Program, Augusta University, Georgia Cancer Center, Augusta, GA 30912, USA. Correspondence and requests for materials should be addressed to D.I.G. (email: dgabrilovich@wistar.org)

Cross-presentation of antigens is a major characteristic of dendritic cells (DC) allowing these cells to induce immune responses. Following uptake, exogenous antigens are internalized into phagosomes (lysosomes) or endosomes[1, 2] and then follow two main processing pathways: cytosolic and vacuolar. The cytosolic pathway involves the transfer of exogenous antigens from the lysosomes into the cytosol for proteasomal degradation. Similar to direct presentation, this pathway is dependent on the transporter for antigen presentation (TAP), and peptide loading on MHC class I molecules occurs either in the endoplasmic reticulum (ER) or in the lumen of endosomes or phagosomes. In contrast, the vacuolar pathway is largely TAP-independent and includes direct loading of peptides onto MHC class I molecules that recycle through the endocytic compartments by peptide exchange. The use of each pathway depends on the type of antigen and the mechanism of its uptake[3]. Proteasome-dependent but TAP-independent mechanism of cross-presentation was also described. It appears to be operational when high doses of soluble antigens are used[4]. Peptide loading in endocytic compartments requires the presence of MHC class I molecules. Therefore it is suggested that MHC class I molecules can be stored in recycling endosomes[5].

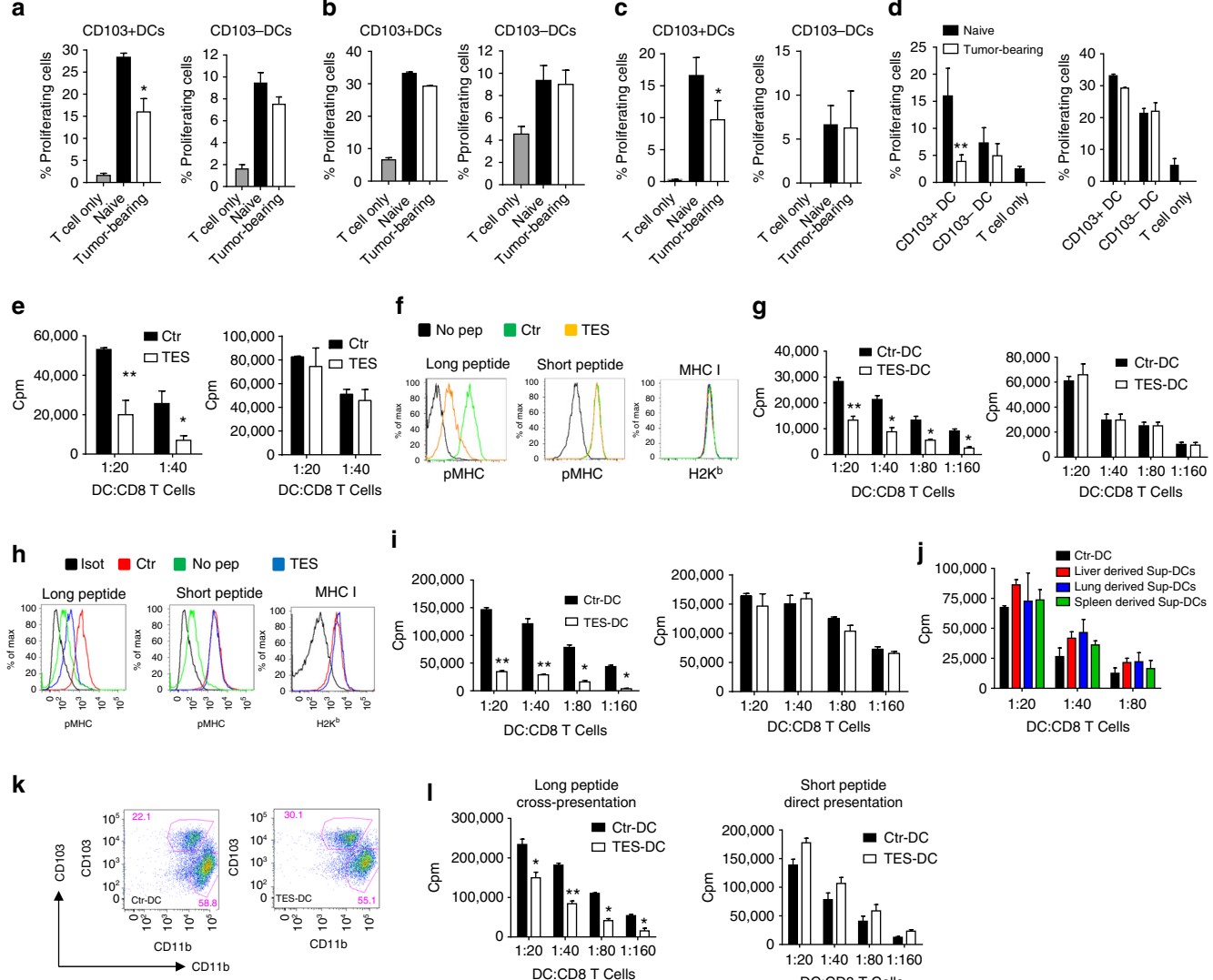

**Fig. 1** Defective cross-presentation by DC in cancer. **a–d** Proliferation of CD8+ T cells measured at 72 h by CFSE dilution. DCs were obtained from dLNs of naïve or TB mice 3 days after immunization with HPV DNA (**a–c**) or OVA-LV vaccines (**d**). CD8+ T cells were isolated from mice immunized with HPV E6/E7 vaccine (**a–c**) or from OT1 transgenic mice (**d**). **a** Antigen-specific CD8+ T cells; **b** allogeneic T cells; **c** antigen-specific CD8+ T cell proliferation after stimulation with DCs isolated from dLNs of naïve or TB mice and then loaded ex vivo with HPV-derived peptides. **d** proliferation of OT1 CD8+ T cells (left panel) and allogenic T cells (right panel) after stimulation with DCs isolated from LN of mice immunized with OVA-LV. Each group included three mice. **e** proliferation of OT1 CD8+ T cells after stimulation with DC generated with GM-CSF, treated with TES and loaded with OVA protein (left panel) or short peptides (SIINFKEL) (right panel). Typical example of 7 experiments is shown. **f, h** Expression of pMHC and MHC class I (H2K^b) by DC generated with GM-CSF (**f**) or FLT3 ligand (**h**) treated with TES for 48 h and then loaded with OVA-derived long or short peptides. Typical example of seven experiments is shown. **g, i** Proliferation of OT1 CD8+ T cells after stimulation with DC generated with GM-CSF (**g**) or FLT3 ligand (**i**) (left panels: DCs loaded with long peptide, right panels: DCs loaded with short peptide). Typical example of seven experiments is shown. **j** Proliferation of OT1 CD8+ T cells after stimulation with DC treated with indicated supernatants for 48 h and loaded with OVA-derived long peptides. **k, l** DCs generated in vitro from BM progenitors with GM-CSF and FLT3L were treated with TES for 48 h. **k** Typical phenotype of cells. **l** Proliferation of antigen-specific CD8+ T cells after stimulation with sorted CD103+DCs. Cumulative results of three experiments are shown. **e–l** Proliferation was measured by ³[H]-thymidine uptake in triplicates. In all experiments mean and SD are shown *p < 0.05, **p < 0.01 in unpaired two-tailed Student's t test between compared groups

Cross-presentation is critically important for antitumor immunity. Antitumor responses were abrogated in Batf3-deficient mice lacking DCs with cross-presenting activity[6]. DCs are present in tumor microenvironment[7–10] and it is known that DC from tumor-bearing (TB) mice are able to cross-present tumor antigen to cytotoxic T lymphocytes (CTL)[11–14]. The clinical success of cancer immunotherapy relies on effective cross-presentation of tumor antigens by DCs[15, 16]. During tumor progression DC have access to large amounts of tumor antigens[17, 18]. The tumor milieu contains soluble mediators such as type I IFN, and endogenous "danger signals" (DNA, HMGB1, S100), which are able to activate DC. Taken together, all these factors induce DC differentiation and activation. However, this does not result in the development of potent antitumor immune responses. Moreover, the induction of strong immune responses to cancer vaccines is a difficult task, even in patients with a relatively small tumor burden. Tumor microenvironment can inhibit immune responses via multiple mechanisms. Among them is the defect in the ability of tumor-associated DC to cross-present antigens[19–22]. However, the mechanism of defective cross-presentation remained unknown.

Lipid droplets or lipid bodies (LB) were implicated in cross-presentation via their association with ER-resident 47 kDa immune-related GTPase, Igtp (Irgm3)[23]. LBs are neutral lipid storage organelles present in all eukaryotic cells. LBs were implicated in the regulation of immune responses via prostaglandins and leukotrienes and, possibly, in interferon responses (reviewed in ref. [24]). Under physiological conditions in most cells, LBs are relatively small with a diameter ranging from 0.1 to 0.2 $\mu m$[25]. In the tumor microenvironment, DCs accumulate larger LB and these have been implicated in defective cross-presentation[22, 26]. This concept was recently confirmed and expanded by different groups[27–31]. Accumulation of lipids in DCs, from TB hosts, is mediated via upregulation of the scavenger receptor (Msr1 or CD204)[26]. This receptor binds various acetylated and oxidized (ox-)lipids[32]. Another mechanism may involve accumulation of ox-lipids as a result of tumor-associated ER stress response[31]. Our previous study showed that LBs do not co-localize with any cellular compartment associated with cross-presentation or with peptide–MHC-I complexes (pMHC) and treatment of DC with IFN-γ did not rescue the defect of cross-presentation, despite the substantial upregulation of MHC-I[22]. Thus, the mechanism regulating cross-presentation by LB in cancer has remained unknown.

Here, we report our unexpected findings demonstrating that LBs accumulating in cancer associated DCs contain oxidatively truncated (ox-tr) electrophilic lipids, which covalently interact with major stress-induced peptide chaperone heat shock protein 70 (HSP70). This prevents trafficking of pMHC from the phagosome/lysosome to the cell surface. As a result, DCs are not able to effectively stimulate antigen-specific T cells.

## Results

### DC cross-presentation in cancer.
It is widely accepted that the efficacy of cancer immune therapy depends on the ability of DCs to cross-present antigens. However, how strong DC cross-presentation is in cancer remained unclear. To assess the effect of tumors on DC cross-presentation in vivo we used HPV vaccine that delivers protein via transcutaneous electroporation[33]. Similar vaccine is currently in clinical trials[34]. Mice with relatively small subcutaneous tumors (TC-1) were vaccinated with HPV DNA vaccine. Three days after vaccination, draining lymph nodes (dLN) were collected and two populations of DCs were sorted: CD11c$^+$CD11b$^-$MHCclassII$^+$CD103$^+$ (CD103$^+$DCs) and CD11c$^+$CD11b$^+$MHCclassII$^+$CD103$^-$ (CD103$^-$DCs). These DCs were used for stimulation of syngeneic E6/E7 HPV-specific CD8$^+$

T cells (generated by vaccination of naïve mice with the same vaccine) or allogeneic T cells obtained from Balb/c mice. As expected, CD103$^+$ DCs from tumor-free mice had potent ability to cross-present antigens to CD8$^+$ T cells. It was lower in DCs from TB mice (Fig. 1a). CD103$^+$ and CD103$^-$ DCs from control and TB mice had similar activity in stimulation of allogeneic T cells (Fig. 1b). This indicated that overall functional activity of DCs was not compromised. In non-immunized mice, cross-presentation was assessed by loading CD103$^+$ DCs with HPV-derived long peptides with subsequent stimulation of antigen-specific CD8$^+$ T cells. CD103$^+$DCs from LN of TB mice had reduced cross-presentation as compared to CD103$^+$ DCs from control mice (Fig. 1c).

In the other model, mice were immunized with a lentiviral vector expressing full-length OVA. Three days after s.c. vaccination two populations of DCs were sorted from dLNs and used for stimulation of OVA-specific CD8$^+$ T cells from OT1 transgenic mice and allogeneic T cells from Balb/c mice. CD103$^+$ DCs in TB mice had dramatically reduced ability to cross-present OVA antigens as compared to naïve mice. In contrast, the ability to stimulate allogeneic T cells was not affected. No differences were observed in CD103$^-$ DCs (Fig. 1d). Thus, DCs from TB mice had substantial defect in cross-presentation.

To evaluate the mechanisms regulating DC cross-presentation by tumor-derived factors (TDF), we generated DCs in vitro from bone marrow (BM) progenitors using two methods allowing for the generation of fully differentiated DCs: 5-day culture with GM-CSF or 6-day culture with FLT3L. Consistent with a previous report[35], cells generated in the presence of GM-CSF were comprised of macrophages (MΦ) and CD135$^+$ DCs (Supplementary Fig. 1a). These DCs also express CD24 and we used this marker to sort in our experiments with FLT3L culture (Supplementary Fig. 1). Tumor explant supernatants (TES) from different tumor cell lines (EL-4 lymphoma or LLC lung carcinoma) were added for an additional 48 h. In already differentiated DCs, TES did not change the phenotype and proportion of these populations or the expression of MHC or co-stimulatory molecules (Supplementary Fig. 1a, b). TES also did not affect the stimulation of allogeneic CD8$^+$ T cells (Supplementary Fig. 1c).

To evaluate the ability of DCs to cross-present antigens, DCs were loaded with OVA protein or OVA-derived long peptide (Pam)2-KMFVESIINFEKL, which require processing and cross-presentation[22] or OVA-derived short peptide (SIINFEKL) that directly binds to MHC class I H2K$^b$ (pMHC). The expression of SIINFEKL/H-2K$^b$ complexes (pMHC) on the cell surface was evaluated using the 25-D1.16 antibody. We found that TES affected the ability of DC to present OVA-derived antigens to peptide-specific OT-1 CD8$^+$ T cells without impairing the direct presentation of short peptide (Fig. 1e). In both, GM-CSF (GM-DCs) or FLT3L (FLT3L-DCs) generated DCs, TES did not affect the expression of pMHC after loading with short peptide. In contrast, it dramatically reduced the expression of pMHC on DCs after loading with long peptide (Fig. 1f, h). TES did not affect the ability of DC to present short peptide (Fig. 1e, g–i, right panel), whereas cross-presentation of OVA-derived antigens (Fig. 1e, left panel) or long peptides was significantly impaired (Fig. 1g, h, left panel). Supernatants prepared from single cell suspensions of liver, lung, or spleens had no effect on cross-presentation (Fig. 1j), indicating that only TDF are able to drive defects in cross-presentation in DCs.

GM-DCs loaded with long peptide after incubation with TES generated a significantly lower antigen-specific response in vivo after sub-cutaneous injection into naïve mice than GM-DCs not treated with TES (Supplementary Fig. 1d). To better characterize the effect of TES in FLT3L-DCs, we evaluated the populations of

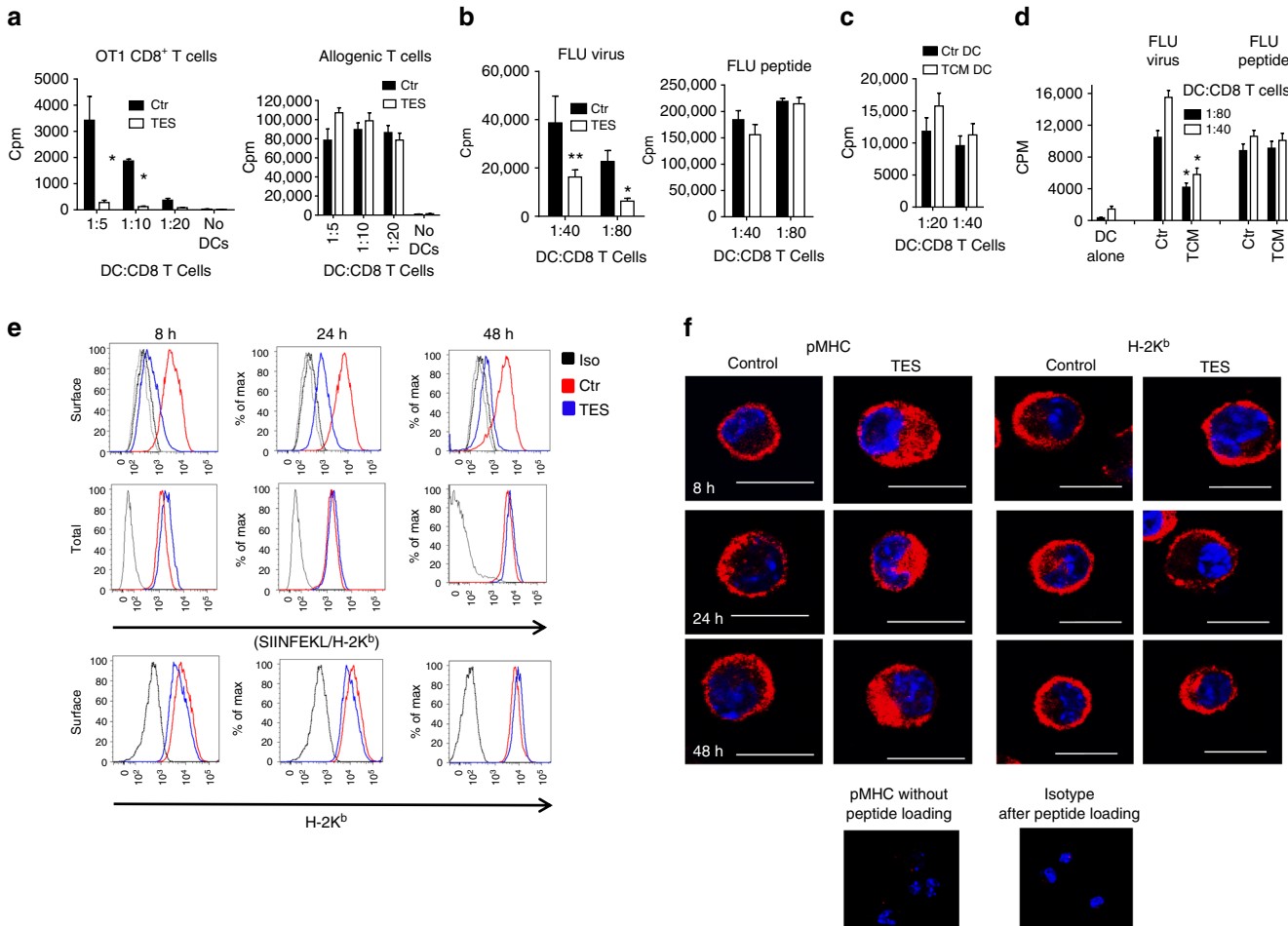

**Fig. 2** Effect of TES on DC function and pMHC distribution. **a** Proliferation of OT1 CD8+ T cells (ratio 1:5) after stimulation with DC treated with TES for 48 h and loaded with apoptotic tumor cells (EG7). **b** Proliferation of HA transgenic CD8+ T cells stimulated with DC generated with GM-CSF, treated with TES for 48 h and then infected with FLU or loaded with FLU short peptide. *$p < 0.05$, **$p < 0.01$ (unpaired two-tailed Student's $t$ test) between control and TES-treated DC. Three experiments with the same results were performed. **c** Proliferation of human allogeneic CD8+ T cells, stimulated with monocyte derived DC treated with TCM. Typical example of one of three performed experiments is shown. **d** Proliferation of CD8+ T cells (ratio 1:40) of HLA-A2.1+ healthy donor stimulated with monocyte derived DC treated with TCM and infected with FLU or loaded with specific FLU peptide. *$p < 0.05$ (unpaired two-tailed Student's $t$ test). Typical example of three experiments is shown. Bars represent standard deviation (SD). Statistical analysis by unpaired two-tailed Student's $t$ test with significance determined at *$p < 0.05$ and **$p < 0.01$. **e, f** Analysis of the expression of H2K^b and pMHC in DC treated with TES and loaded with OVA-derived long peptide. Typical example of flow cytometry (**e**) and confocal microscopy (**f**) is shown. Four experiments with the same results were performed. Scale bars = 10 μm

CD24^high^CD11b^low/−^DCs and CD11b^high^CD24^low/−^DCs. TES caused significant decrease in cross-presentation of long peptide but did not affect direct presentation of short peptide by these cells (Supplementary Fig. 1e, f). Since GM-DCs resemble migratory DCs[35], in most experiments we used DCs generated with GM-CSF.

We directly compared the effect of TES on cross-presenting ability of DCs and MΦ generated with GM-CSF (Supplementary Fig. 2a). TES caused substantial decrease in pMHC expression and stimulation of specific CD8+ T cell proliferation of both DCs and MΦ after loading with long peptide (Supplementary Fig. 2b, c). However, expression of pMHC and stimulation of CD8+ T cells after cross-presentation by DCs was almost tenfold higher than that by MΦ, which supports critical role of DCs in cross-presentation. No effect of TES on direct presentation was seen (Supplementary Fig. 2d).

CD103+ DCs are most potent cross-presenting DCs. We generated B220^−^CD11c^+^CD11b^low^Sirpα^low/neg^CD103^+^DCs as previously described[36] and incubated with TES (Fig. 1k). Sorted

CD103+ DCs treated with TES had defect in cross-presentation but not in a direct presentation (Fig. 1l).

To determine that the defects in cross-presentation were not limited only to OVA, we evaluated the ability of GM-DCs to cross-present antigens associated with apoptotic tumor cells and derived from influenza virus (FLU). DCs were co-cultured with apoptotic EG7 cells expressing full-length OVA protein and then tested in their ability to stimulate OT-1 CD8+ T cells. DCs exposed to TES had substantially lower ability to stimulate antigen-specific CD8+ T cells than control DCs. Importantly, those DCs potently stimulated allogenic T cells (Fig. 2a). TES from 4T1 tumor decreased the ability of GM-DCs generated from BALB/c mice to cross-present FLU to CD8+ transgenic T cells specific for HA-derived peptide. In contrast, TES did not affect the presentation of short H2D^d^-matched FLU peptide to these T cells (Fig. 2b).

Tumor-conditioned medium (TCM) obtained from the human SK-MEL melanoma cell line did not affect the ability of human DCs generated from CD14+ monocytes to stimulate allogeneic

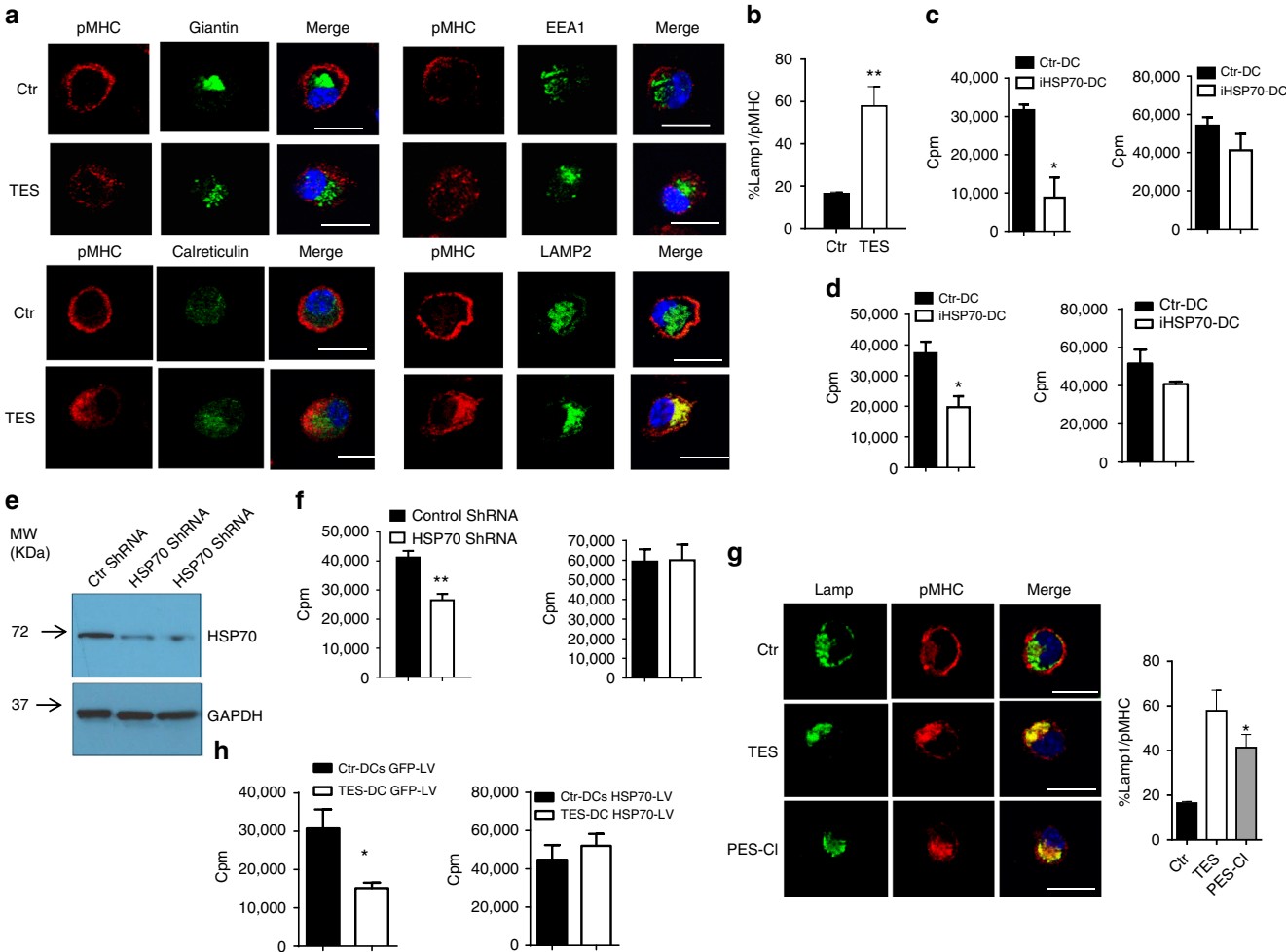

**Fig. 3** The role of HSP70 in regulation of cross-presentation by DC. **a** Co-localization of pMHC with indicated intracellular compartments of DCs treated with TES: calreticulin—ER, EEA1—early endosomes, LAMP2—late endosomes/lysosomes, giantin-Golgi. Five experiments were performed. At least 50 cells were evaluated in each experiment. Typical examples are shown. Scale bar = 10 μm. **b** Proportion of DCs with co-localization of pMHC and Lamp2, representative of three performed independently. At least 50 cells were counted in each experiment. Bars represent standard deviation (SD). Statistical analysis by unpaired two-tailed Student's $t$ test with significance determined at *$p < 0.05$ and **$p < 0.01$. **c**, **d** Proliferation (in triplicates) of OT1 CD8[+] T cells, stimulated with DCs (at 1:20 ratio). DCs were treated with TES and HSP70 inhibitors PES-Cl (**c**) or PET 16 (**d**) and loaded with OVA long peptide (left panels) or short peptide (right panels). Typical examples of three performed experiments are shown. *$p < 0.05$, **$p < 0.01$ (unpaired two-tailed Student's $t$ test) from control untreated DC. **e** Knock down HSP70 in DC, using HSP70 shRNA expressing lentiviral vectors. **f** Proliferation in triplicate of OT1 CD8[+] T cells (ratio 1:20) stimulated with control shRNA or HSP70 shRNA transduced DCs loaded with OVA long (left panel) or short peptides (right panel). DC: T cells ratio 1:20 is shown. Mean and SD in one of two experiments with the same results are shown. *$p < 0.05$ (unpaired two-tailed Student's $t$ test) between control shRNA and HSP70 shRNA DC. **g** pMHC and Lamp1 in DC treated with HSP70 inhibitor PES-Cl or TES analyzed by confocal microscopy. On the right—proportion of cells with co-localization of pMHC. At least 50 cells were evaluated in each of four performed experiments. Typical examples are shown. Scale bar = 10 μm. **h** Proliferation of OT1 CD8[+] T cells (ratio at 1:20) after stimulation with DCs infected with GFP-LV or Hsp70-LV and loaded with OVA long peptides. Proliferation was measured by [3][H]-thymidine uptake in triplicates. *$p < 0.05$. Bars represent standard deviation (SD). Statistical analysis by unpaired two-tailed Student's $t$ test with significance determined at *$p < 0.05$ and **$p < 0.01$

T cells (Fig. 2c). However, DC treated with TCM for 48 h and infected with FLU had reduced stimulation of autologous CD8[+] T cells. This effect was absent when DCs were loaded with short HLA-A2.1 matching FLU peptide (Fig. 2d).

Taken together, these data indicate that TDF cause profound defects in cross-presentation by differentiated DCs, which was not, however, associated with a decrease in MHC class I or co-stimulatory molecule expression, or inhibition of binding of short MHC class I matched peptide.

In an attempt to understand the mechanism of inhibition of antigen-cross presentation by DCs in cancer, we evaluated the distribution of pMHC in DCs. Long OVA peptides generated a high density of pMHC which could be detectable. In control DCs, pMHC were seen on the cell surface within 8 h after loading with

long peptide. It was still clearly detectable after 48 h (Fig. 2e). In DCs treated with TES, expression of pMHC on the surface was substantially lower. It was evident at the earliest time point and further decreased with time. When cells were fixed and permeabilized, the differences in the amount of pMHC between control and TES-treated DCs were not detectable suggesting that the total amount of complexes inside the cell was not affected. DCs exposed to TES retained a control level of H2K[b] expression (Fig. 2e). Confocal microscopy was used to confirm those observations. In contrast to control DCs, where pMHC complexes were localized on the surface of the cells, in TES-treated DC, pMHC complexes were largely intracellular. No differences were seen in the localization of total MHC class I (Fig. 2f).

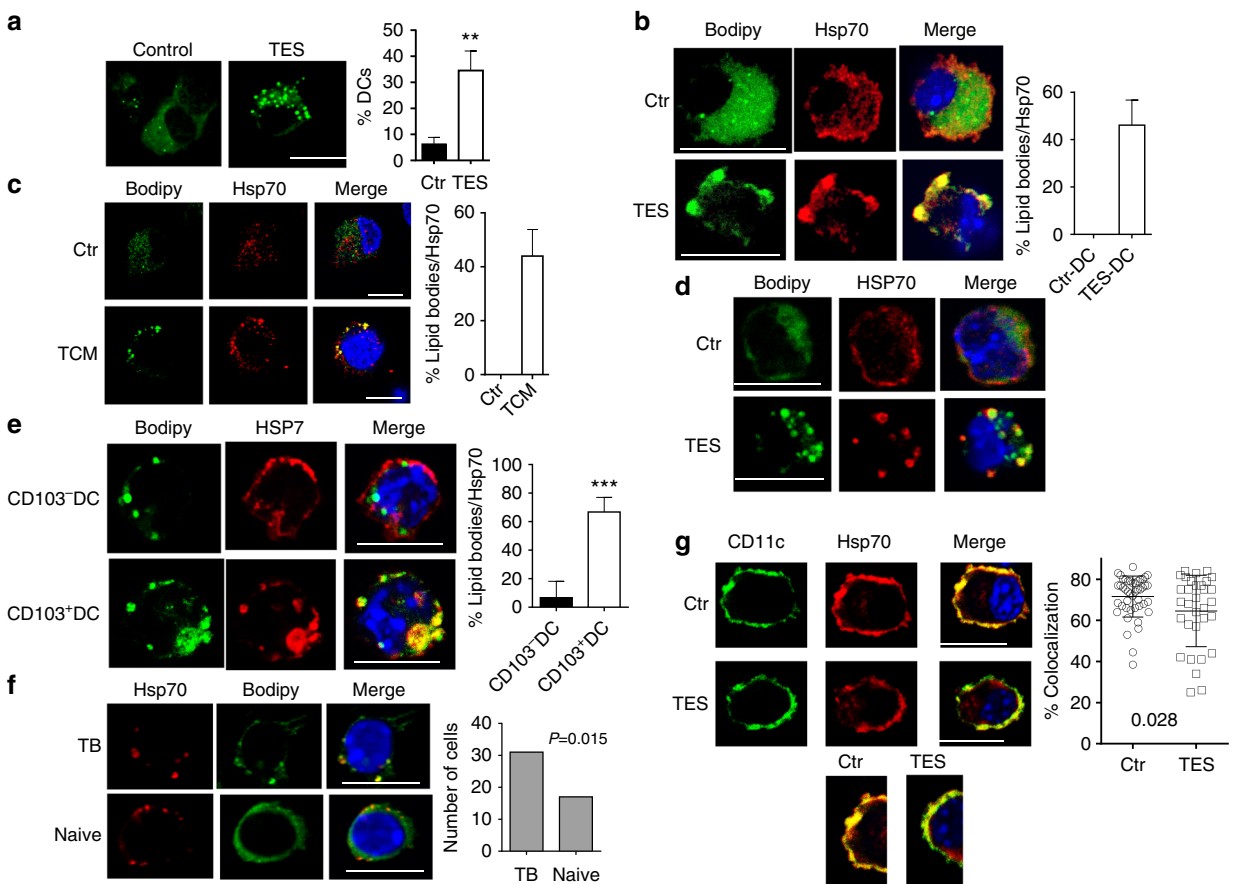

**Fig. 4** Interaction between LB and HSP70 in DC. **a** Confocal microscopy analysis of LB in DCs treated with TES. Typical example is shown. Scale bar = 10 μm. Proportion of DCs containing more than three large LB (minimal size 0.4 μm) was calculated. For each sample at least 100 cells were counted. The results from three samples are shown. **p < 0.01. **b, c** Confocal microscopy of LB and HSP70 in DC treated with TES (**b**) or in human monocyte derived DC treated with TCM (**c**). Scale bar = 10 μm. Four different experiments were performed and typical example is shown. The proportion of DCs with co-localization of LB and HSP70 **b, c** was calculated from the total cells containing LBs. At least 50 cells in each sample were counted. Four samples were evaluated **d**. DCs were generated from HPC using GM-CSF and FLT3L and then exposed to TES. **d** Confocal microscopy of sorted CD103+ DCs loaded with long peptide. Scale bars = 50 μm. **e** CD103+ and CD103− DCs were sorted from dLNs of TB mice and evaluated. On the left—typical example of staining. On the right—proportion of co-localization of lipid bodies with HSP70. Scale bar = 10 μm. Four independent experiments were performed. ***p < 0.001 in unpaired two-tailed Student's t test. **f** Co-localization of HSP70 with LB in DC sorted from draining LN of LLC bearing mice or LN of naïve mice. Left—typical examples of staining. Scale bar = 1 μm. Right—number of cells with co-localization LB and HSP70 out of 100 cells counted. Cumulative results obtained from three mice. p value in Fisher's rank test. **g** Co-localization of HSP70 and CD11c in DC treated with TES. Left—typical example of staining. Scale bar = 10 μm. Right—% of co-localization for HSP70 and CD11c in gated part of the cell calculated using Leica Software. At least 40 cells were counted in each sample, total four samples

Next, we evaluated co-localization of pMHC complexes with various intracellular compartments. After loading of control DC with long peptide, pMHC were observed largely on the cell surface and did not co-localize with early endosomes, Golgi, or ER. Surprisingly, in more than 60% of all TES-treated DCs, pMHC co-localized with LAMP2 positive lysosomes/late endosomes (Fig. 3a, b). H-2K$^b$ also co-localized with late endosomes/lysosomes in TES-exposed DC, albeit to a lesser extent than pMHC, and the expression of H2K$^b$ on the cell surface was not affected (Supplementary Fig. 3a). Co-localization of pMHC with lysosomes was also observed in TES-treated CD103+ DCs (Supplementary Fig. 3b). Thus, defect in cross-presentation was associated with inability of pMHC to traffic to the cell surface and their accumulated in lysosomal/late endosomal compartment.

**Regulation of cross-presentation by HSP70 and its association with LB**. We next sought to determine the mechanism underlying impaired trafficking of pMHC. We investigated heat shock

proteins 70 and 90 (HSP70 and HSP90), as these have been previously implicated in antigen presentation and cross-presentation by DCs[37–40]. Selective HSP70 inhibitor PES-Cl, which blocks substrate binding domain[41], caused a decrease in DC cross-presentation, but not in direct presentation (Fig. 3c). PES-Cl did not affect DC viability or major co-stimulatory molecules (Supplementary Fig. 3c, d). PET-16 inhibitor, which binds to an allosterically regulated domain of HSP70[42], had the same effect (Fig. 3d). In contrast, selective inhibition of HSP90 by 17-N-Allylamino-17-demethoxygeldanamycin (17AAG) did not affect cross-presentation by DCs. Moreover, at lower concentrations, there was an upregulation of cross-presentation (Supplementary Fig. 4a). Since it is known that 17AAG upregulates HSP70[43], this data further supports a potential role of HSP70 in cross-presentation. To better evaluate the specific nature of HSP70 involvement into cross-presentation, we downregulated HSP70 in DCs generated from BM progenitors using HSP70 shRNA lentiviral vector (Fig. 3e, raw data on WB can be found in Supplementary Fig. 5) and then loaded with long or

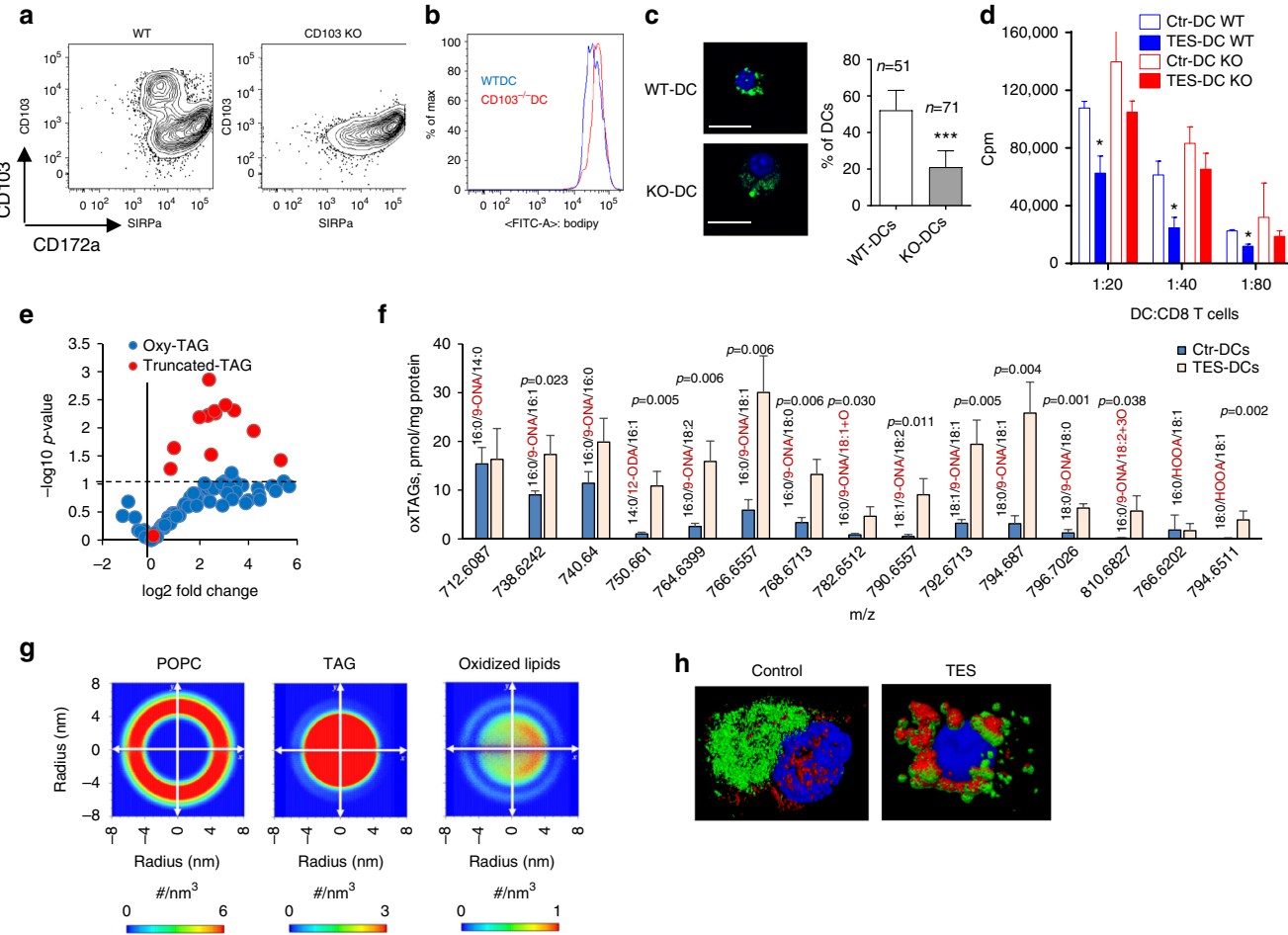

**Fig. 5** Lipid accumulation and models interaction of HSP70 with oxidized LB. **a–d** DCs were generated from bone marrow HPC of wild-type (WT) and CD103$^{-/-}$ (KO) mice using 9 day culture with GM-CSF and FLT3L followed by 48 h incubation with TES. **a** Phenotype of cells demonstrating lack of CD103 expression in CD11c$^+$ DCs. **b** Lipid level in DCs after staining with BODIPY in DC treated with TES. **c** LB (BODIPY staining) in sorted TES-treated CD11c $^+$CD172$^-$ DCs analyzed by confocal microscopy. Typical example of staining (blue—DAPI, green—BODIPY) and the proportion of DCs with large LB (>0.4 µm) calculated per cell are shown. The number of cells counted is shown on the graph. Scale bar = 50 µm. **d** Cross-presentation of long OVA-derived peptide by DCs. *$p < 0.05$, ***$p < 0.001$ (unpaired two-tailed Student's $t$-test) from WT. **e** Volcano plot showing lipidomics analysis of oxidized TAG (singly- and doubly-oxygenated TAG, blue) and oxidatively truncated TAG (red) in TES-DC vs. Ctr-DC Horizontal dashed-line corresponds to the significance level of $p = 0.05$ with points above the line having $p < 0.05$ and points below the line having $p > 0.05$. **f** The amount of various ox-tr-TAG molecular species in control DC and DC treated with TES. **g** CG-MD simulations of density of POPC, non-oxidized TAG and oxidized lipids in the LB along the $z$-axis of the simulation box. **h** 3D confocal analysis of co-localization between HSP70 and LB in BM derived DC treated with TES. Green—LB, red—HSP70

short OVA peptides. Partial silencing of HSP70 did not affect direct binding, but caused a decrease in the ability of DC to cross-present long peptide (Fig. 3f). Inhibition of HSP70 in long-peptide loaded DCs prevented re-distribution of pMHC complexes to the cell surface and increased co-localization of pMHCs with lysosomes, thus recapitulating the effect of TES (Fig. 3g). To test whether overexpression of HSP70 can rescue TES inducible defect in DC cross-presentation, DCs were transduced with lentiviral vectors expressing either GFP (control) or *hsp70* (Hspa1a) and then exposed to TES followed by loading with long peptide. TES caused a decrease in cross-presentation by DCs transduced with control vector. In sharp contrast, DCs transduced with *hsp70* lentivirus retained their ability to cross-present antigen (Fig. 3h) without affecting expression of MHC class I (Supplementary Fig. 4b). Thus, HSP70 is involved in regulation of DC cross-presentation in cancer.

**Accumulation of lipids in DC and HSP70**. We next sought to determine how HSP70 might contribute to the defective cross-

presentation. Consistent with previous observations, TES caused an accumulation of large (>0.4 µm) LB in GM-DCs (Fig. 4a) as well as in CD24$^+$ DCs generated in the presence of FLT3L (Supplementary Fig. 4c, d). LB co-localized with HSP70 in mouse (Fig. 4b) and human (Fig. 4c) DCs was generated with GM-CSF and treated with TES or TCM. In contrast, no co-localization was detected between small LB present in control DC and HSP70 (Fig. 4b, c). TES also caused accumulation of large LB co-localized with HSP70 in CD103$^+$ DCs (Fig. 4d). Co-localization between LB and HSP70 was found in CD103$^+$ but not CD103$^-$ DCs isolated from draining LN in TB mice (Fig. 4e). In vivo, CD11c$^+$MHCII$^+$F4/80$^-$ DCs isolated from the dLN of LLC TB mice had a greater amount of lipids than DC from non-draining LN from the same mice (Supplementary Fig. 4e) and CD103$^+$ DCs had higher amount of lipids than CD103$^-$ DCs (Supplementary Fig. 4f). A higher proportion of CD11c$^+$MHCII$^+$DCs in dLN had co-localization of HSP70 with LB than DCs from naïve mice (Fig. 4f).

Using confocal microscopy, we investigated a localization of HSP70 in different cellular compartments of DCs generated

in vitro with GM-CSF. CD11c was used as a marker of cellular membrane and Lamp1 as marker of lysosomes/late endosomes. After loading of DCs with long peptide, HSP70 was readily detectable on the cell membrane co-localizing with CD11c. In contrast, DCs treated with TES had significantly lower co-localization with CD11c (Fig. 4g). TES significantly decreased the presence of HSP70 in lysosomes (Supplementary Fig. 4g), suggesting that the re-distribution of HSP70 could be involved in changes in cross-presentation. These results prompted us to further investigate possible role of LB-HSP70 interaction in cross-presentation.

**Lipid accumulation, redox lipidomics, and model of interaction of LB with HSP70.** We previously implicated scavenger receptor CD204 in lipid accumulation by DCs[26]. Since CD103+ DCs accumulate more LB we investigated the possible role of CD103 in this process by using CD103 deficient DCs generated

in vitro from CD103 KO mice (Fig. 5a). WT and CD103−/− DCs accumulated lipids equally well (Fig. 5b). However, CD103−/− DCs had significantly lower proportion of DCs with large LB (Fig. 5c). In contrast to WT DCs, TES-treated CD103−/− DCs had no defect in cross-presentation (Fig. 5d).

We found that the expression of genes involved in lipogenesis was not increased in CD103+ DCs exposed to TES or isolated from LN of TB mice (Supplementary Fig. 6a, b). Two genes: *dgat2* and *dgat1* were upregulated in vitro and in vivo, respectively. The enzymes encoded by these genes are involved in formation of LB directly from fatty acid (FA) that are picked up by cells and are found on the surface of LB[44]. This supports the conclusion that lipid uptake is likely to be a major factor regulating formation of LB in DCs in TB hosts.

Lipidomics and redox-lipidomics analysis of lipids present in DC exposed to TES revealed 250 molecular species of non-oxidized (non-ox)lipids of which 165 were represented by

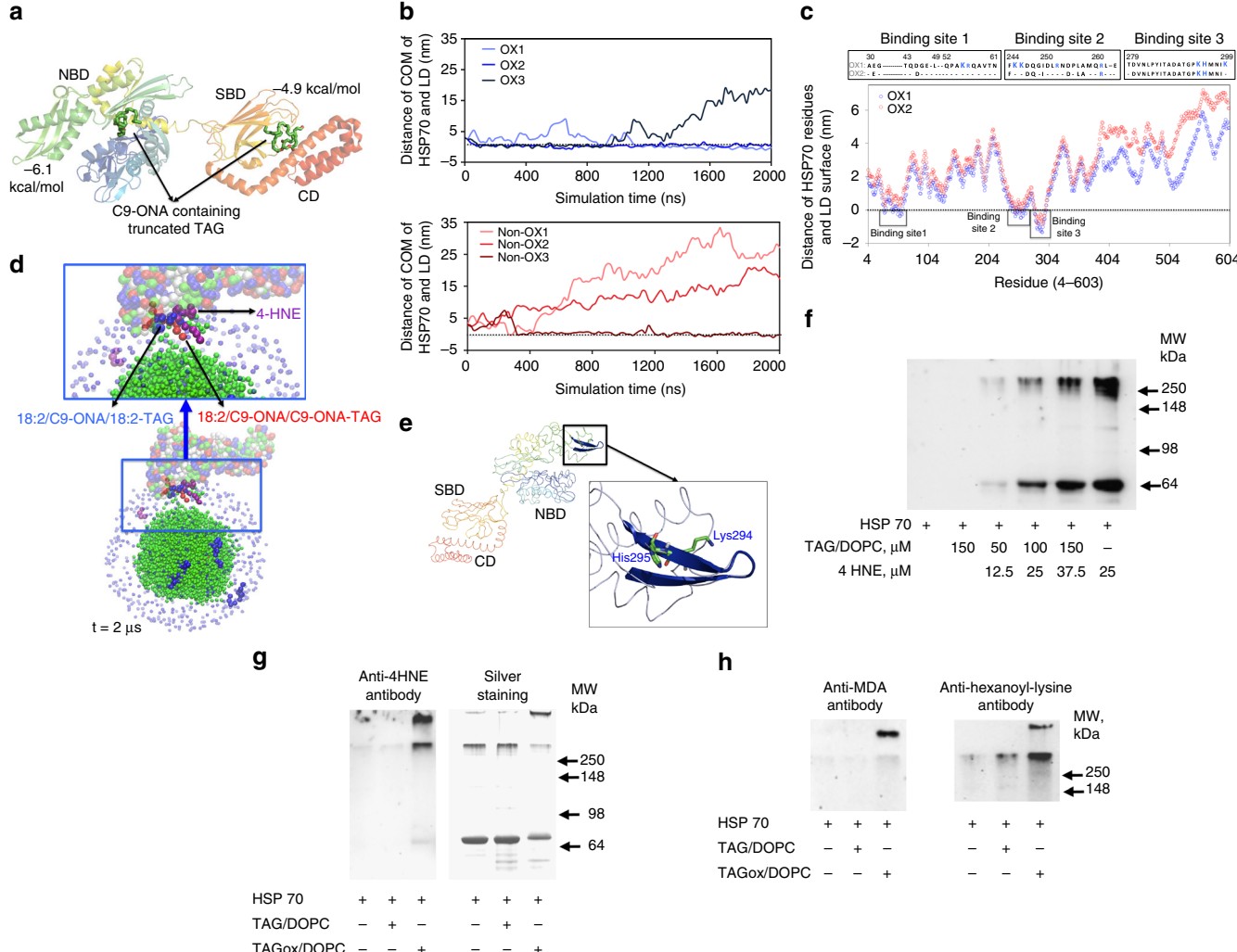

**Fig. 6** Interaction of HSP70 with oxidized LB. **a** Molecular docking showing affinity interaction of the oxygenated truncated TAG with N-terminal ATPase domain (NAD) as compared with the substrate binding domain (SBD) or C-terminal domain (CD). **b** HSP70 interacts with oxTAG containing LB in all three CG-MD simulations during 2 μs runs, in which the initial orientations of HSP70 were varied as compared with the interactions in only one out of three runs with non-oxTAG containing LB. **c** Two out of three CGMD simulations (OX1 and OX2) showed strong interactions of HSP70 and oxTAG containing LB until the end resulting in the similar final orientations (total simualtion 2 μs). **d** A typical final configuration of CGMD simulations shows that HSP70 interacts mainly with oxygenated products in ox-tr-TAG containing LB. **e** The main binding site responsible for the anchoring of HSP70 to the LB surface contains a sheet-turn-sheet motif and a positively charged Lys294—likely involved in the covalent interactions with oxTAG. Bars represent standard deviation (SD). Statistical analysis by unpaired two-tailed Student's *t* test with significance determined at *$p < 0.05$ and **$p < 0.01$. **f–h** Detection of 4-HNE adducts **f**, **g** or MDA- and HEL-adducts **h** formed after incubation of recombinant HSP70 with 4-HNE in the presence of TAG/DOPC (**f**), or AA containing TAG/DOPC oxidized by MPO (**g**, **h**)

phospholipids of all major classes. DCs also contained 74 molecular species of triacylglycerides (TAG) and 11 molecular species of cholesterol esters (CE). The total amount of TAG in DC treated with TES was elevated as compared to control DCs (Supplementary Fig. 7a). This effect was mostly due to the major TAG molecular species containing monounsaturated oleic acid (C18:1) and polyunsaturated linoleic acid (LA) (C18:2). (Supplementary Fig. 7b–d). No significant differences between control and TES-treated DC were observed for CE (Supplementary Fig. 7e–g).

We did not find oxidatively modified phospholipids of any type in DCs. In contrast, we detected several types of oxidation products in TAG and CE. In both classes of these neutral lipids, we identified oxidation products of two types—long chain oxygenated species (ox-) and oxidatively truncated (ox-tr-) species formed via oxidative degradation of the side chain. Notably, higher contents were found for ox-tr electrophilic TAG (Fig. 5e, f). The most prominently elevated ox-tr-TAG in TES exposed DCs were ox-tr molecular species formed by cleavage of LA—9-oxo-nonanoic acid (9-ONA) and arachidonic acid (AA)—5-hydroxy-8-oxo-6-octenoic acid (HOOA) (Supplementary Fig. 8a). An increased content of mono- and doubly oxygenated TAG in DCs in the presence of TES compared to control DC was observed (Supplementary Fig. 8b–d). In CE, the levels of ox-long chain molecular species containing epoxy-groups as well as ox-tr-LA, were significantly increased (Supplementary Fig. 8e–h). Overall, the LC–MS data clearly demonstrated the selective lipid oxidation of the major components of LB—TAG and CE, whereas membrane phospholipids remained refractive to the oxidation process.

LB contain a monolayer of phospholipids on their surface[45]. These phospholipids represent only a minor fraction of the total phospholipids in DCs. Therefore we evaluated the presence of ox-lipids directly in LB isolated by gradient centrifugation. Redox lipidomics showed that phosphatidylcholine (PC) was the major component of LB and was represented by LA and AA acyls (Supplementary Fig. 9a, b). Fragmentation analysis identified oxidation product as hydroxy-derivative of AA (hydroxy-eicosatetraenoate, HETE)—a relatively stable compound non-reactive towards proteins[46]. Neither control DC nor TES-DC contained ox-tr-phospholipid species readily interacting with proteins. In contrast, analysis of TAGs in LB revealed significant amounts of ox-tr-TAGs derivatives (Supplementary Fig. 9c). The contents of these ox-tr-TAGs products were higher in TES-DC than in control DC and exceeded many-fold the contents of hydroxy-products in PC. Based on these results, we conclude that ox-tr-electrophilic species capable of interacting with proteins were present exclusively in LB neutral lipids.

To gain molecular insights into the possible anchoring of HSP70 to LB, we performed computational modeling of their interactions using ox- and non-ox-TAG. Using coarse-grained molecular dynamics (CG-MD) simulations, we examined the motional behavior and preferred localizations of ox-tr-TAG within LB. CG-MD demonstrated the higher likelihood of ox-tr-TAG vs. non-ox-TAG to translocate from the LB hydrophobic core into the phospholipid monolayer on the LB surface (Fig. 5g) thus making them accessible for interactions with multiple cytosolic proteins, including HSP70. To directly test this hypothesis, we performed an experiment with DCs using confocal microscopy with 3D analysis. LB present in control DCs did not co-localize with HSP70. In DC exposed to TES, HSP70 was found localized on the surface of LB (Fig. 5h).

The molecular docking analysis showed that ox-tr-TAG strongly interacted with N-terminal ATPase domain of HSP70 with the binding energy of −6.1 kcal/mol in contrast to a much weaker binding to the substrate binding domain or C-terminal domain (binding energy of −4.9 kcal/mol) (Fig. 6a). We utilized CG-MD simulations to explore the interactions of HSP70 with LB containing ox-tr-TAG and non-ox-TAG. Since the full-length structure of human HSP70 has not been published, we employed the structure of the *Escherichia coli* HSP70 orthologue DnaK with 46.5% identity and 26.3% similarity, which represent a high degree of homology[47]. We assessed the distance of center of mass of HSP70 to the surface of LB during 2 μs (Fig. 6b). For non-ox-TAG-LB, only one trial revealed binding with HSP70. In contrast, HSP70 interaction with ox-tr-TAG-LB was observed in all three cases tested. By the end of the simulations, HSP70 remained tightly bound to the surface of ox-tr-TAG-LB in two runs, and was positioned in the same orientations (Fig. 6c).

The typical final configurations of HSP70 anchored to ox-TAG-LB are shown in Fig. 6d. To investigate the sequence conservation between the *E. coli* and human HSP70 proteins, these two sequences were aligned. We observed that Sites 1 and 2 contain two conserved lysine residues (lys55 and lys245) that are predicted to be responsible for the covalent immobilization of HSP70 on the ox-TAG-LB surface[48]. Therefore, HSP70 can be anchored on the surface of ox-TAG-LB through a two-step process: (i) the initial interactions governed by long-range non-bonded forces including electrostatic and hydrophobic forces and (ii) short-range chemical interactions leading to lipidation of HSP70. Site 3 does not include any conserved lysine residues, however, it comprises 10 hydrophobic amino acid residues. We assigned 15 Å penetration of this beta hairpin like binding site (Fig. 6e) into the ox-TAG-LB based on the high level of hydrophobicity of this site likely leading to non-covalent stabilization of HSP70 on the surface of ox-TAG-LB.

To verify chemical interactions between HSP70 and ox-tr-TAG in simple biochemical systems we used PAGE shift-gel assay. Oxidative truncation of PUFAs in TAG leads to the formation of different electrophilic products (e.g., aldehydes including 4-hydroxy-2-nonenal (4-HNE) and malonyl-dialdehyde (MDA)) (Supplementary Fig. 10a). 4-HNE is a highly electrophilic molecule which forms adducts with nucleophilic thiol (−SH) or amino (−NH2) groups present in amino acids such as Cys, His, and Lys, via Michael addition (Supplementary Fig. 10b). HNE can also modify protein structure through Schiff base formation with lysyl residues, leading to pyrrole formation (Supplementary Fig. 10c, d). We employed an antibody specific for protein adducts with 4-HNE to assess the potential interaction of recombinant HSP70 with 4-HNE added to a mixture of non-oxidizable TAG with 1,2-dioleoyl-sn-glycero-3-phosphocholine (DOPC) at a ratio 50:1 mimicking the composition of LB. 4-HNE formed covalent adducts with HSP70 in a concentration-dependent manner (Fig. 6f). Further, we assessed covalent association of HSP70 with ox-tr-TAG generated by pre-incubation of AA and LA-containing TAG with MPO. LC–MS analysis showed that incubation of TAG(AAA)/DOPC or TAG (LLL)/DOPC with MPO/H2O2/NaCl resulted in the formation of a variety of ox-TAG, including ox-tr species. Western blot analysis detected 4-HNE adducts in the monomeric form of HSP70 (70 kD) as well as in high molecular weight protein aggregates present in the gel (Fig. 6g). Aggregated high molecular weight protein oligomers are characteristic products formed as a result of protein crosslinking by bifunctional lipid peroxidation products (Supplementary Fig. 10e). High molecular weight aggregates formed after interaction of HSP70 with ox-TAG contained epitopes recognized by antibodies against MDA-adducts and hexanoyl-lysine adduct (HEL) (Fig. 6h), which are also products of oxidative modification of LA or AA. Epitopes recognizable by antibodies against HEL adduct were also found after interaction of HSP70 with LA-containing ox-TAG (Supplementary Fig. 10f). These combined data support the premise that

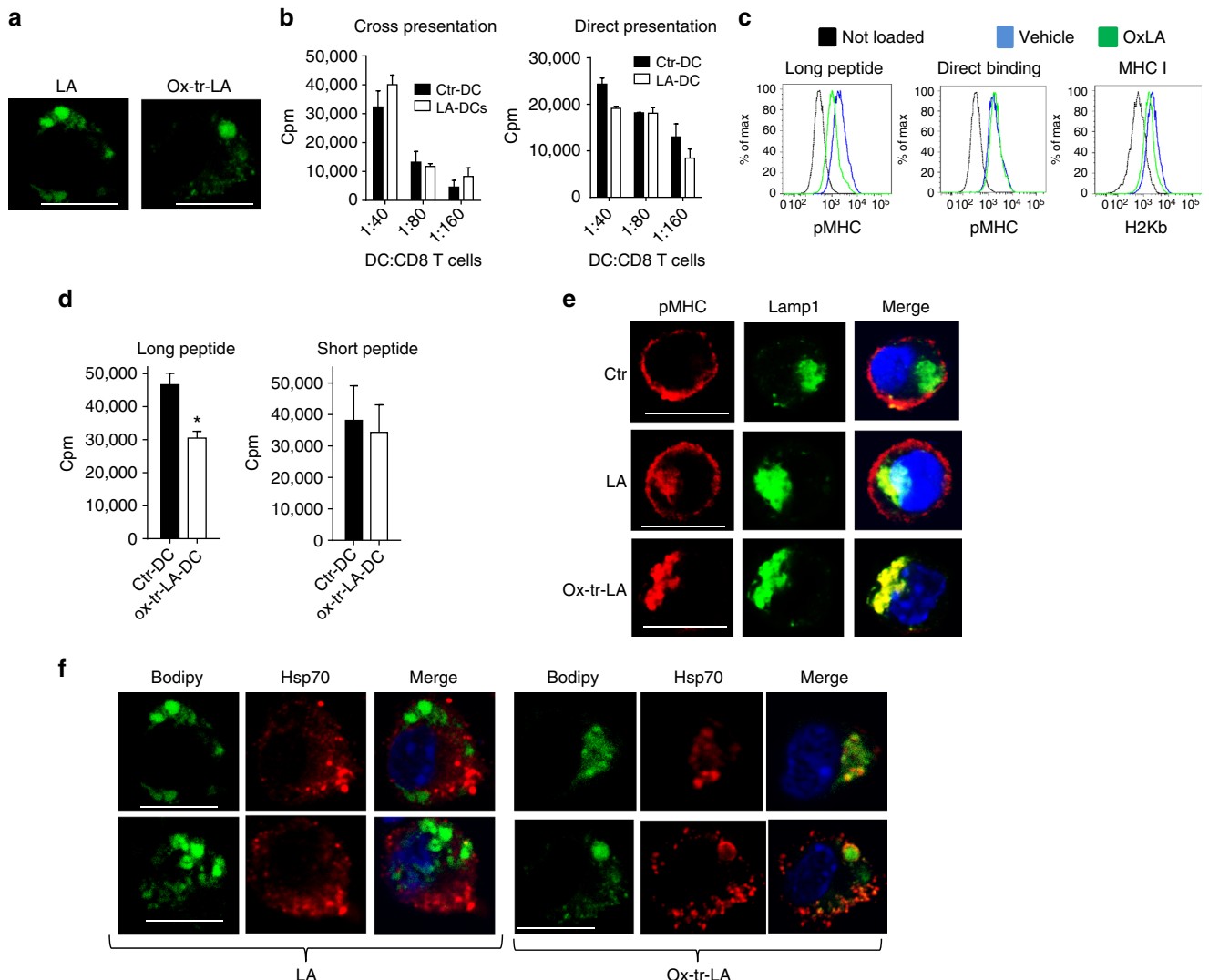

**Fig. 7** Interaction of HSP70 with oxidized LB directly affect cross-presentation in DC. **a** Confocal microscopy analysis of LB in BM derived DC treated with LA or ox-tr-LA. Scale bar = 10 μm. **b** Proliferation of CD8[+] T cells (ratio at 1:40) stimulated with DC treated with 100 μM LA and then loaded with OVA-derived long (left panel) or short peptides (right panel). Typical example of three performed experiments is shown. Proliferation was measured in triplicates, mean and SD are shown. **c** Expression of pMHC measured by flow cytometry in BM derived DC treated with 100 μM ox-tr-LA and then loaded with OVA long or short peptides. Representative of four independent experiments performed. **d** Proliferation of OT1 T cells (ratio at 1:20), stimulated with BM derived DC treated with ox-tr-LA and then loaded with OVA long (left panel) or short peptides (right panel). Proliferation was assessed by [3][H]-thymidine incorporation in triplicates. Three experiments with the same results were performed. Mean and SD are shown. *$p < 0.05$ from control. **e** Co-localization between pMHC and Lamp1 in BM derived DC treated with LA or ox-LA and loaded with OVA long peptide. Scale bar = 10 μm. Typical example of staining is shown. Bars represent standard deviation (SD). Statistical analysis by unpaired two-tailed Student's $t$ test with significance determined at *$p < 0.05$ and **$p < 0.01$. **f** Co-localization between LB and HSP70 in BM-derived DC treated with 100 μM LA or ox-tr-LA and loaded with OVA long peptide. Scale bar = 10 μm. Two typical examples of staining are shown

LB containing ox-tr-TAG are able to covalently bind to HSP70 and preclude HSP70 interaction with pMHC, thus affecting pMHC trafficking to cell surface.

**LB enriched with ox-tr-lipids binds to HSP70 and prevented cross-presentation**. To test the possibility that oxidized lipids bind to HSP70 and inhibit its function, we prepared ox-tr-LA in vitro using MPO plus a pro-oxidant system (Fe + ascorbate). Various oxygenated metabolites of LA were determined by LS-MS (Supplementary Table 1). DC were loaded with non-ox-LA and ox-tr-LA for 4 h in serum-free medium (SFM) at concentrations that did not affect DC viability or phenotype (Supplementary Fig. 11) but resulted in the accumulation of large LB (Fig. 7a).

Loading of DC with LA did not cause the defect in cross-presentation (Fig. 7b). In contrast, DC loaded with ox-tr-LA had significantly impaired cross-presentation of long peptide without affecting direct binding and presentation (Fig. 7c, d). Loading of DC with LA did not affect pMHC expression on the surface and did not cause co-localization of pMHC with Lamp1 positive lysosomes (although some co-localization was detectable). In contrast, ox-tr-LA disrupted pMHC localization on the surface and instead lead to its co-localization with lysosomes (Fig. 7e). LB generated in DC after loading with LA did not co-localize with HSP70, whereas LB developed after loading with ox-tr-LA demonstrated co-localization in most of the cells (Fig. 7f).

We investigated the effect of anti-oxidant α-tocopherol (vitiamin E, Vit. E) on formation of LB and cross-presentation

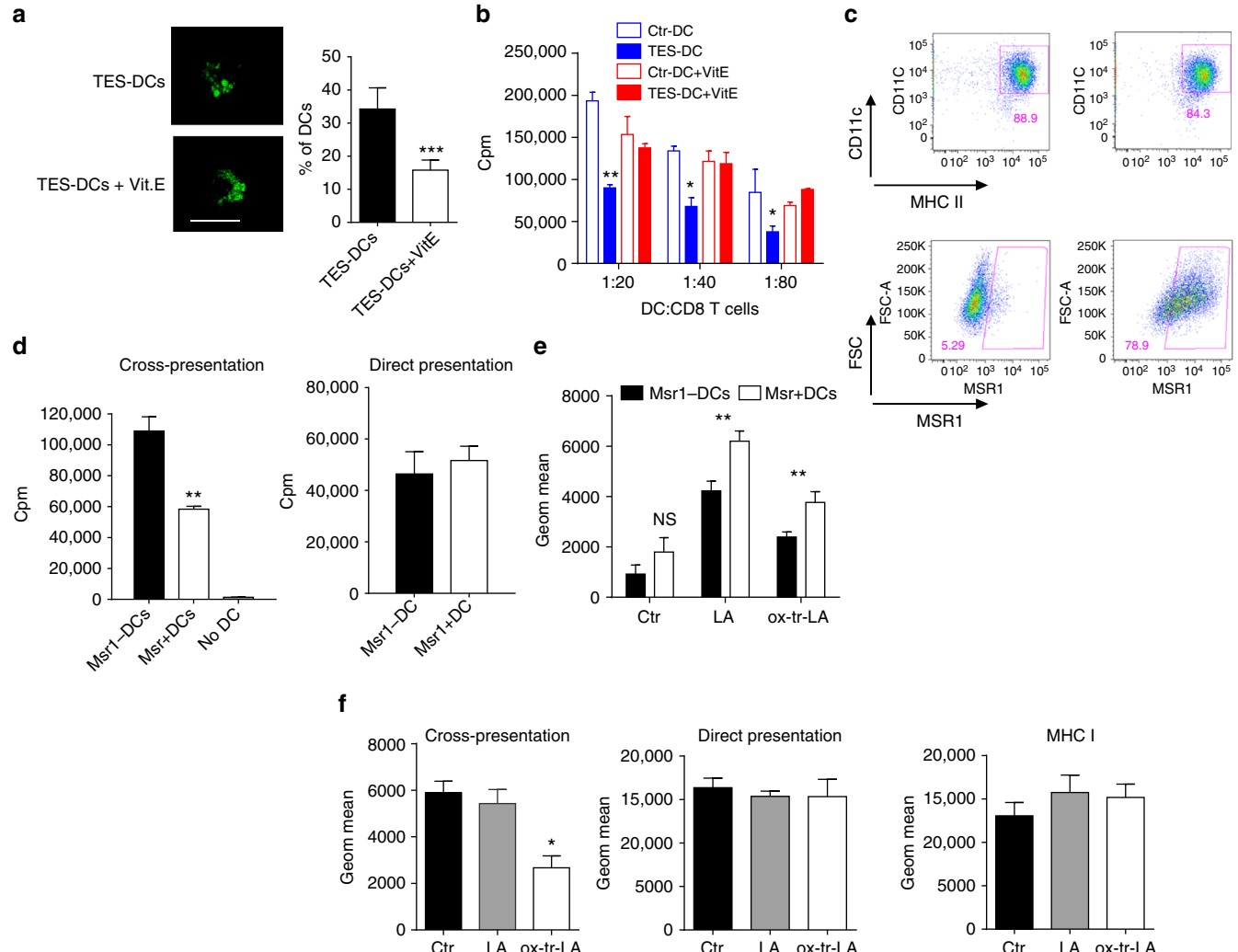

**Fig. 8** Effect of vitamin E and overexpression of Msr1 on DC cross-presentation. **a, b** DCs were generated from HPCs with GM-CSF and Flt3L and exposed to TES. Cells were pre-incubated with Vit. E and after 8 h TES was added. **a** The number of large LB per cell (CD11c⁺ CD103⁺ CD172a⁻ DCs). Typical example of staining with BODIPY and the proportion of DCs with the presence of large LB are shown. Scale bar = 50 μm. **b** Cross-presentation of long OVA-derived peptide by DCs. *p < 0.05, **p < 0.01, ***p < 0.001. **c** Typical example of the overexpression of Msr1 in DC after infection with Msr1 lentiviral vector. **d** Proliferation of OT1 T cells, stimulated with Msr1⁻ or Msr1⁺ DC, loaded with OVA long or short peptides. DC: T cell ratio 1:40 is shown. Proliferation was measured in triplicates. Mean and SD are shown. Two experiments with the same results were performed. *p < 0.05. **e** Accumulation of lipids in Msr1⁺ DC and Msr1⁻ DC treated for 48 h with 100 nM LA or ox-tr-LA an then stained with Bodipy. Two experiments with the same results were performed. **f** Expression of pMHC and H2Kᵇ in DC treated with 100 μM LA or ox-tr-LA and then loaded with long (cross-presentation) or short (direct presentation) OVA-derived peptide. Three experiments were performed. Mean and SD are shown. *p < 0.05. Bars represent standard deviation (SD). Statistical analysis by unpaired two-tailed Student's t test with significance determined at *p < 0.05, **p < 0.01 and ***p < 0.001

by generating DCs with TES in the presence of Vit. E (100 μM). Vit. E did not affect accumulation of lipids but decreased the number of large LB (Fig. 8a) and abrogated defects in cross-presentation caused by TES (Fig. 8b). By acting as a radical scavenger, Vit. E cannot affect the already formed products of lipid peroxidation. However, it can prevent the generation of peroxidized lipids, including ox-tr-lipids. This may explain the protective effects of Vit. E against TES induced defects in cross-presentation.

Our previous studies demonstrated that the accumulation of ox-lipids in DC in cancer was largely due to upregulation of one of the scavenger receptors, Msr1 (CD204)[22, 26]. To establish the direct link between lipid accumulation and cross-presentation we overexpressed Msr1 in DC using a lentiviral construct (Fig. 8c). Overexpression of Msr1 did not affect expression of MHC class I, class II, or co-stimulatory molecules on DC. It did not affect

direct presentation but significantly reduced cross-presentation (Fig. 8d). Overexpression of Msr1 facilitated uptake of both LA and ox-tr-LA by DC in SFM (Fig. 8e). However, only ox-tr-LA caused defects in antigen cross-presentation. No changes in the direct binding and presentation or MHC class I expression were observed (Fig. 8f).

## Discussion

Cross-presentation by DCs is critically important for antitumor immunity. Even when T cells with pre-determined specificity (TCR-T cells or CAR-T cells) are used, the immune responses to shared antigens depend on the ability of DC to cross-present tumor antigens. We and others have previously demonstrated that cross-presentation in DC from TB mice is impaired[19–22]. Here, we have identified a mechanism of this impaired cross-presentation.

In recent years accumulation of lipids was implicated in the defective cross-presentation by tumor-associated DCs[22, 26–30]. However, the mechanism of this phenomenon remained unclear. Morphologically, LB accumulated in tumor-associated DC were much larger than LB present in control DC. However, these LB did not co-localize with pMHC or cellular compartments[22]. The remarkable feature of lipids found in tumor-associated DC, in contrast to control DC, was accumulation of ox-TAG and CE, the main components of LB[25]. The predominance of ox-tr-electrophilic species suggests that their reactivity towards nucleophilic amino-acid target proteins is a likely mechanism of oxidative lipidation of a target critical for cross presentation. Since DCs have relatively ineffective machinery for oxidation of intracellular lipids we hypothesized that these cells can pick up ox-lipids from tumor microenvironment. Overexpression of CD204 in control DC reduced cross-presentation. Although overexpression of CD204 facilitated uptake of both non-ox- and ox-lipids, only ox-lipids inhibited cross-presentation further supporting important role of ox-lipids in defective cross-presentation in cancer. More recently it has been suggested that lipids may accumulate in tumor DC as the result of ER stress[31]. We did not find evidence of increased lipogenesis induced by TES. CD103[+] DCs accumulate LB much more than CD103[−] cells. It is possible that CD103 may contribute to the formation of LB in DCs since lack of CD103 resulted in decrease in the number of LB in DCs. CD103 is integrin alpha E. This integrin has not been implicated in formation of LB. However, other intergins (primarily αvβ3 and αvβ5) were shown to be important for this process[49, 50].

We turned our attention to chaperone HSP70, which is has been shown to be involved in antigen presentation[37, 38]. HSP70 can also stabilize lysosomes by binding to an endolysosomal anionic phospholipid bis(monoacylglycero)phosphate, an essential co-factor for lysosomal sphingomyelin metabolism[51]. The inhibition of HSP70, either pharmacologically or genetically, blocked cross-presentation of long peptides by DC. In control mouse and human DC, HSP70 did not co-localize with LB. In striking contrast, in mouse and human DC exposed to TDF or isolated from TB mice, LB strongly co-localized with HSP70. We hypothesized that in tumor DC, LB could bind to HSP70 and thus prevent effective cross-presentation. This hypothesis was supported by the fact that inhibition of HSP70 in control DC not only blocked cross-presentation but caused co-localization of pMHC with lysosomes, similar to the effect of TES. It suggested that the nature of LB could determine their ability to bind HSP70 and block cross-presentation. To test this hypothesis directly, we loaded DC with either non-ox-LA or ox-LA. Since LA is used for synthesis of TAG and other lipid species loading of DC with LA caused accumulation of large LB. However, non-ox-LA did not affect cross-presentation, trafficking of pMHC to cell surface, and did not cause co-localization of LB and HSP70. In contrast, loading of DC with ox-LA inhibited cross-presentation, which was associated with accumulation of pMHC in lysosomes and co-localization of LB with HSP70. Thus, accumulation of LB containing non-oxidized lipids was not sufficient for the inhibition of the cross-presentation and interaction between LB and HSP70. We hypothesized that the presence of polar electrophilic oxygen-containing groups in ox-lipids or ox-tr-TAG could be responsible for the interaction between LB and HSP70 in tumor DC. Addition of hydrophilic oxygen-containing functionalities during peroxidation of polyunsaturated FA may markedly change the hydrophobic/hydrophilic balance of the latter, which may dramatically affect the structural organization of lipid-enriched domains of membranes and LB. Computational CG-MD simulation supported this hypothesis. The computational analysis was directly confirmed in experiments with 3D confocal microscopy and further supported by molecular docking simulations. HSP70 has been deemed to be an attractive target for cancer therapy, given the considerable overexpression of this stress-induced chaperone on cancer vs. normal cells[52]. Our data suggest that HSP70 inhibitors may abrogate cross-presentation by DC, suggesting that caution in the use of these inhibitors should be exercised.

Thus, our data describe a mechanism of inhibition of cross-presentation by DC in cancer mediated by ox-lipids via their interaction with HSP70. Identification of ox-tr-lipids of LBs as essential contributors to inhibition of cross presentation of tumor antigens may suggest new potential targets for therapeutic regulation of cross-presentation.

## Methods

**Human cell and mouse models.** Human studies were approved by The Wistar Institute IRB. Peripheral blood was collected from seven healthy volunteers after obtaining informed consent. Animal experiments were approved by The Wistar Institute Animal Care and Use Committee. Human experiments were approved by Wistar Institutional Review Board and informed consent was obtained from all subjects. Balb/c or C57BL/6 mice (female, 4–6 week old, about 20 mice used) were obtained from Charles River, OT-I TCR-transgenic mice (C57Bl/6-Tg(TCRaTCRb) 1100mjb) (female, 4–6 week old, about 10 mice used), B6.129S2(C)-Itgaetm1Cmp/J mice (female, 4–6 week old, about 4 mice used), B10.Cg-H2d Tg(TcraCl4,TcrbCl4) 1Shrm/ShrmJ (female, 4–6 week old, about 4 mice used) were purchased from Jackson Laboratory.

**Reagents and cell lines.** Tumor cell lines including EL4 lymphoma, LLC (Lewis Lung Carcinoma), and B16F10 melanoma were maintained in DMEM medium supplemented with 10% fetal bovine serum (FBS, Sigma-Aldrich, St. Louis, MO) at 37 °C, 5% $CO_2$. Tumors were injected subcutaneously (s.c.) at $5 \times 10^5$ cells per mouse. Tumor cell lines were tested for mycoplasm contamination by using Universal Mycoplasma detection kit (ATCC). SIINFEKL peptide and control peptide RAHYNIVTF were obtained from American Peptide Company (Vista, CA), (Pam)2−KMFVESIINFEKL peptide (derived from OVA) and (Pam)2-KMFVKVPRNQDWL (derived from gp100) were obtained from DBA Synthetic biomolecules (San Diego). Recombinant mouse GM-CSF was obtained from Invitrogen. Recombinant human GM-CSF and IL-4 were obtained from Peprotech. Anti-mouse CD11c conjugated beads were purchased from Miltenyi, and used for DC purification. FITC or APC conjugated anti mouse CD11c antibodies (Cat. no. 45-0114-82; clone: N418; 0.2 µg/million cells) were obtained from eBiosciences. APC or PE conjugated anti mouse SIINFEKL-H2Kb complex antibody (Cat. no. 12-5743-82. Clone: 25-D1.16, 0.1 µg/million cells), un-conjugated anti mouse SIINFEKL-H2Kb complex antibody (Cat. no. Clone: 25-D1.16), PE conjugated mouse F4/80 (Cat. no. 12-4801-82. clone: BM8. 0.1 µg/million cells) were purchased from eBioscience. Percp5.5 conjugated mouse MHC class I (Cat. no. 562831. H2Kb, clone: AF6.88.5, 0.2 µg/million cells) antibody, PE-Cy7 conjugated mouse MHC class II (Cat. no. 116419. IAb; clone: AF120.1, 0.05 µg/million cells) antibody, CD172a (Cat. no. 144007; clone: P84; 0.5 µg/million cells) were purchased from BioLegend, BV421 conjugated mouse CD11b (Cat. no. 101236; clone: M1/70; 0.02 µg/million cells) and CD103 (Cat. no. 121422; clone: 2E7, 0.2 µg/million cells) were purchased from BioLegend, BV421 conjugated mouse CD80 (Cat. no. 562611; clone: 16.10A1; 0.1 µg/million cells) antibodies, FITC or PE conjugated mouse CD86 (Cat.no. 553692; clone: GL1; 0.1 µg/million cells) antibodies, APC conjugated amouse CD40 (Cat.no. 553790; clone: 3/23; 0.1 µg/million cells) antibody were all purchased from BD Bioscience. BODIPY lipid dye 493/503 was obtained from Invitrogen. The antibodies for detection different cellular compartments including EEA1 antibody (marker for early endosome. Cat. no. ab2900; 1 µg/ml), Giantin antibody (marker for Golgi complex. Cat. no. ab24586; 1:100), Lamp2 antibody (marker for lysosome Cat. no. ab25339; 2.5 µg/ml) and calreticulin antibody (marker for ER. Cat. no. 92516; 0.8 µg/ml) were obtained from Abcam. HSP70 antibody (Cat. no. 610607; 0.5 µg/ml), anti-mouse lamp1 (Cat. no. 553792, clone: 1D4B; 1.25 µg/ml) were purchased from BD Biosciences. Aqua live, Alexa Fluor 488, Alexa Fluor 647 conjugated anti-mouse (Cat. no. A21202; 5 µg/ml), anti-rabbit (Cat. no. A21245; 5 µg/ml), and anti-rat (Cat. no. A11006; 5 µg/ml) secondary antibodies were obtained from Invitrogen. Dapi was purchased from life Technology.

**Preparation of TES and TCM.** Tumor explant supernatants were prepared from excised non-ulcerated tumors ~1.5 cm in diameter. Tumors were minced into pieces <3 mm in diameter, washed with PBS 1× and resuspended in RPMI 1640 supplemented with 2 mM L-glutamine, 200 U/ml penicillin plus 50 µg/ml streptomycin, 55 µM β-mercaptoethanol (Gibco) and 10% FBS. The cell free supernatant were collected after 16–18 h of incubation at 37 °C and kept at −80 °C. Tissue conditioned media (TCM) were prepared from excised liver, lung and spleen, by following the procedure above.

**Cell phenotype and lipid contents by flow cytometry**. DC were incubated with FC-block (BD Biosciences) for 10 min and surface staining was performed at 4 °C for 15 min. Cells were run on LSRII flow cytometer (BD Biosciences) and data were analyzed by FlowJo (Tristar). For lipid staining, cells were re-suspended in 500 µl of Bodipy 493/503 at 0.25 µg/ml in PBS. Cells were stained for 15 min at room temperature in the dark, then washed twice, re-suspended in PBS and run immediately on LSRII. At least 10,000 cells were collected for subsequent analysis.

**Generation of DCs**. Mouse DCs were generated from enriched BM hematopoietic progenitor cells (HPCs) with 10 ng/ml of GM-CSF. Briefly, HPCs were isolated from mouse BM by using Lineage depletion kit (Miltenyi), according to manufacturer's instructions. Cells were seeded at 50,000 cell/ml in 24 well plates and GM-CSF (10 ng/ml) was added to the culture at day 0 and day 3. At day 5, cells were collected and CD11c positive cells were isolated by using anti-CD11c conjugated beads and then were cultured for additional 48 h in the presence of fresh medium containing 20% v/v TES and 10 ng/ml of GM-CSF. In some experiments DC were generated from HPCs with 200 ng/ml of FLT3L (Peprotech). In some experiments DC were generated with GM-CSF and FLT3L. The cytokines were added again at day 3 and cells were used at day 6 or day 7.

Human DC were generated from human CD14 positive monocytes. Briefly, HLA-A2.1 positive PBMCs were obtained from donors' buffy coat by ficoll-Paque (GE Healthcare) gradient centrifugation and CD14 positive monocytes were isolated from PBMC by using anti-human CD14 conjugated beads (Miltenyi), following the manufacturer's instructions. Monocytes were cultured for 5 days with 25 ng/ml of rhGM-CSF and 25 ng/ml of rhIL-4. Cytokines were added again at day 3. On day 5, media was replaced and 20% tumor-conditioned media (derived from SK-MEL melanoma cell lines) was added. 48 h later, the non-adherent and loosely adherent cells were collected.

**Isolation of DC from LN**. LN were digested for 30 min at 37 °C with collagenase A (0.5 mg/ml; Sigma Aldrich), Dnase I (0.2 mg/ml, Roche), diluted in HBSS with $Ca^{2+}/Mg^{2+}$ and 20 mM EDTA (Invitrogen) was added 5 min at room temperature to stop the reaction. Single suspensions were prepared and then DC were stained and sorted on BD FACS Aria (BD Biosciences).

**Cross-presentation in vivo**. *HPV-DNA vaccine*: 6–8-week-old C57Bl/6 mice were implanted with 50,000 TC-1 HPV16 E6/E7 expressing tumor cells (gift from Dr. Yvonne Paterson) subcutaneously in both flanks. Naïve or TB mice were immunized once with 25 µg (in water) of HPV16 E6E7 plasmid (pGX3001) by intramuscular injection (IM) into the Tibialis anterior (TA) muscle followed by electroporation (EP) using the CELLECTRA -3P adaptive constant current electroporation device (Inovio Pharmaceuticals, Inc.). Two 0.1 A constant current pulses (52 ms in length) were delivered with a 1 s delay between pulses. Draining LNs were collected and DCs isolated 3 days after immunization. E6/E7 specific CD8[+] T cells used as responders were isolated by using EasySep Mouse CD8[+] T Cell Enrichment Kit (STEMCELL) from spleen of naïve mice, immunized twice with HPV E6E7 plasmid as above and labeled with 0.5 µM of CFSE (Biolend), following manufactures' instruction.

*OVA-lentiviral vector (LV)*: 6–8-week-old C57Bl/6 mice were implanted with $5 \times 10^5$ LLC subcutaneously. Naïve and tumor bearing mice were immunized once with 1.000.000 TU/mouse of OVA-LV subcutaneously. Draining LNs were collected and DC isolated 3 days after immunization. CFSE labeled OT1 CD8[+] T cells were used as responders.

**Cross-presentation of OVA-derived long peptides**. DCs were loaded for 16–18 h with 100 µg/ml OVA or 5 µg/ml long peptides. CD8 T cells were isolated from spleens of responder mice by using EasySep Mouse CD8[+] T Cell Enrichment Kit (STEMCELL) and then plated at $10^5$ T cells per well. DC and T cells were mixed at different ratios. DC were loaded with 0.5 µg/ml of SIINFEKL for 1 h at 37 °C. In some experiments OVA was delivered in DC through CD205 receptors. Briefly, DC were stained with anti-biotin CD205 (Miltenyi) for 10 min at 4 °C. Then cells were labeled with monoclonal anti-biotin antibodies conjugated to OVA, following the manufacturer's instructions (Ova antigen delivery Reagent, Miltenyi). DC were washed, resuspended in complete RPMI supplemented with GM-CSF and incubated for 8 or 16 h at 37 °C before functional assays. In some experiments, DC were used to stimulate allogeneic CD8[+] T cells isolated from spleen of Balb/c mice. At day 3 3[H]-thymidine was added at 1 µCi per well for an additional 18 h followed by cell harvesting and a radioactivity count on liquid scintillation counter.

**Cross-presentation of peptide derived from FLU**. Mouse DC were generated from enriched HPCs of Balb/c mice, as described above. At day 6 DC were isolated using magnetic beads and infected with FLU (A/Puerto Rico/8/1934(H1N1)) (2000 HAU/10[7] cells) for 2 h at 37 °C or loaded with 0.5 µg/ml of IYSTVASSL short peptide (518–526) (AnaSpec Inc.) for 1 h at 37 °C. Cells were washed twice with complete medium and then mixed at different ratios with HA specific CD8[+] T cells isolated from HA TCR transgenic mice using EasySep Mouse CD8[+] T Cell Enrichment Kit. HLA-A2.1 matched human DC, generated from CD14[+] monocytes were infected at day 6 with FLU (2000 HAU/10[7] cells) for 2 h at 37 °C or loaded with 1 µg/ml of CEF1 (influenza matrix protein M1 (58–66)) (AnaSpec Inc.)

short peptide for 1 h at 37 °C. Cells were washed twice with complete medium and then mixed at different ratios with T cells isolated from HLA-A2.1 matched autologous CD8[+] T cells, using EasySep Human CD8[+] T Cell Enrichment Kit. In some experiments human DC were used to stimulate allogeneic T cells isolated from PBMC of healthy donors. At day 4 3[H]-thymidine was then added at 1 µCi per well for an additional 18 hof incubation, followed by cell harvesting and a radioactivity count on liquid scintillation counter.

**Cross-presentation of apoptotic tumor cells**. DCs were co-cultured with apoptotic tumor cells and then used to stimulate OT1 CD8[+] T cells, isolated as described above. Briefly, EG7 (expressing OVA) tumor cell lines were treated with 2 µM of Doxorubicin (Sigma Aldrich) to induce apoptosis. The cells were collected 24 h later, washed three times and then co-cultured with DCs at ratio 3:1. After 24 h DC were collected, washed and used for assays.

**Treatment with LA and ox-tr-LA**. DC were resuspended in serum-free RPMI containing 100 µM LA or 100 µM ox-LA and 10 ng/ml GM-CSF, and incubated for 4 h at 37 °C. Cells were then washed twice with complete RPMI and loaded with OVA long peptide (5 µg/ml) for 16–18 h or loaded with short peptide (SIINFEKL, 0.5 µg/ml) for 1 h at 37 °C. Cells were then used for assays.

**Confocal microscopy**. Dendritic cells were washed twice with PBS 1×, resuspended in complete RPMI and 50,000 cells were seeded on poly-L-lysine cellware 12 MM round coverslips (Corning) for 45 min at 37 °C. After that time, cells were washed with PBS 1× and were stained for surface markers. Briefly, cells were incubated with FC-block (BD Biosciences) for 10 min, stained with un-conjugated antibodies for 15 min at 4 °C, washed twice with PBS before incubation with fluorochrome associated secondary antibodies. Afterwards, cells were fixed and permeabilized with Fixation & Permeabilization Buffers (BD Biosciences) for 15 min at RT, washed twice with wash buffer (BD Biosciences), and then blocked with PBS containing 5% FBS for 45 min. Cells were incubated with FC-block for 5 min at RT and stained with different primary antibodies, at 4 °C for 16–18 h. Cells were washed three times and incubated with fluorochrome associated antibodies for 45 min at RT. After that time cells were washed three times and then stained with BODIPY, to detect lipid bodies for 15 min at RT. Cells were washed and incubated with DAPI and mounted on slides using Prolong Gold antifade reagent (Life Technology). The cells were imaged with a Leica TCS SP5 laser scanning confocal microscope (Leica Microsystems). Rate of co-localization (%) was calculated by using Leica LASX (Microsystems software). Briefly, it is calculated from the ratio of the area of colocalizing fluorescence signals (Colocalization Area) and the area of the image foreground (Area Foreground). Backgrounds and the thresholds were set up before starting the analysis.

**Generation of LV-MSR1, LV-HSP70, and LV-OVA**. The Msr1 gene was excised from the pCMV6-AC-MSR1-GFP plasmid (Origene) using SnaBI/XhoI and cloned into a SIV-based self-inactivating lentiviral transfer vector [53] downstream of the internal CMV promoter (pGAE-CMV-MSR1-Wpre). The HSP70 gene was excised from the pcDNA3.1D/V5-His-TOPO-HA-HSP70 plasmid using SnaBI/XhoI and cloned into a SIV-based self-inactivating lentiviral transfer vector downstream of the internal CMV promoter (pGAE-CMV-HA-HSP70-Wpre). The transfer vectors pGAE-CMV-GFP-Wpre and pTY2-CMV-OVA-Wpre, the packaging plasmid pAd-SIV3+, and the Vesicular Stomatitis virus envelope G protein (VSV-G) pseudotyping vectors from Indiana serotype (pVSV.GIND), have been previously described [53–55]. The human epithelium kidney 293T Lenti-X cells (Clontech) were maintained in Dulbecco's Modified Eagles medium (DMEM) (Gibco) supplemented with 10% fetal bovine serum (FBS) (HyClone) and 100 units/ml of penicillin–streptomycin–glutamine (PSG) (Gibco). For production of recombinant lentiviral vector (LV), $3.5 \times 106$ Lenti-X cells were seeded on 100 mM diameter Petri dishes and transfected with 12 µg per plate of a plasmid mixture containing transfer vector, packaging plasmid and VSV.G plasmid in a 6:4:2 ratio, using the JetPrime transfection kit (Polyplus Transfection) following the manufacture's recommendations. At 48 and 72 h post transfection, culture supernatants were cleared from cellular debris by low speed centrifugation and passed through a 0.45 µm pore size filter unit (Millipore). Filtered supernatants were concentrated by ultracentrifugation for 2 h at 23.000 RPM on a 20% sucrose cushion. Pelleted vector particles were resuspended in 1× PBS and stored at −80 °C until further use. Each LV-Msr1 stock was titered using a reverse transcriptase (RT) activity assay (Reverse Transcriptase Assay, colorimetric, Roche) and the corresponding transducing units (TU) calculated by comparing the RT activity of each LV-MSR1 and LV-HSP70 stock to the RT activity of LV-GFP stocks with known infectious titers [56].

**Treatment with inhibitors and HSP70 knock down**. DCs were pre-treated 30 min at 37 °C with the following inhibitors: 4 or 6 µM of PES-Cl [PES-Cl=2-(3-chlorophenyl) ethynesulfonamide] or PET-16 [Triphenyl(phenylethynyl)phosphonium bromide] or 25 or 50 µM of HSP90 inhibitor. Then the cells were washed twice and used for assays.

HSP70 shRNA was expressed in the pLKO.1 vector obtained from Open Biosystems (TRCN0000013831). shControl was generated in pLKO.1 vector with target sequence 5′-TTATCGCGCATATCACGCG-3′. All shRNA targeting

sequences (TRCN0000008515, TRCN0000098597) were obtained from Wistar protein expression facility. Lentiviruses were produced by co-transfecting pLKO.1 shRNA expression plasmid packaging vectors pMD2.G and pSPAX2.

DCs were generated as described above. At day 3 cells were resuspended in serum-free medium (SFM) containing lentiviral vectors followed by centrifugation of the plate at 2200 rpm for 45 min at 37 °C. Fresh media supplemented with GM-CSF was then added. Puromycin (1 µg/ml) was added 48 h post infection and cells were collected and used for assays 48 h later. HSP70 downregulation was verified by western blot.

**Western blot.** Cells were lysed in RIPA buffer (Sigma-Aldrich) in presence of protease inhibitor cocktail (Sigma-Aldrich). Whole cell lysates were prepared and subjected to 10% SDS-PAGE and transferred to PVDF membrane. The membranes were probed overnight at 4oC with the antibodies specific for HSP70 (BD Biosciences) and GAPDH (Cell Signaling Technology), Membranes were washed and incubated for 1 h at room temperature with secondary antibody conjugated with peroxidase.

**Quatitative real time PCR.** RNA was extracted using Total RNA Kit I (Omega, Biel/Bienne, Switzerland) according to manufacturer's instructions. DNase digestion was performed cDNA was generated with High-Capacity cDNA Reverse Transcription Kit (Applied Biosystems, Foster City, CA, USA). qRT-PCR was preformed using Power SYBR Green PCR Master Mix (Applied Biosystems) in 96-well. Plates were read with ABI 7900 (Applied Biosystems). Amplifications were carried out with the following primers (mouse): β-actin-f, ggaggggggttgaggtgtt; β-actin-r, gtgtgcacttttattggtctcaa; acaca-f, cagtgctatgctgagattgagg; acaca-r, acacagccagggtcaagtg; acacb-f, agttcgccgattcccagt; acacb-r, atggcctcttcacggttct; dgat1⁻f, tcgtggtatcctgaattggtg; dgat1-r, aggttctctaaaaataaccttgcatt; dgat2-f, gctggtgccctactccaag; dgat2-r, ccagcttggggacagtga; msr1-f, gggagtgtaggcggatca; msr1-r, ggagatgatagtagggtgctctg. scd1-f, ggtgatgttccagaggaggta; scd1-r cgcaagaaggtgctaacga.

**Liquid chromatography and mass-spectral analysis of lipids.** Lipids were extracted by Folch procedure (Folch et al., 1957) with slight modifications, under nitrogen atmosphere, at all steps. Prior LC–ESI-MS analysis, lipid extracts were separated by 1D-HPTLC ((5 × 5 cm TLC plates). Special measures were taken to prevent oxidative modification of lipids during their processing and separation. To bind adventitious transition metals from silica, plates were treated with methanol containing 1 mM EDTA, 100 mM diethylene triamine pentaacetic acid prior to application and separation of lipids by 1D-HPTLC. Then the plates were developed with a solvent system consisting of hexane: diethyl ether: glacial acetic acid (75:15:1 v/v). After the plates were dried with a forced N2 blower to remove the solvent. The lipids were visualized by exposure to iodine vapors and identified by comparison with authentic phospholipid standards. For LC–ESI-MS analysis of lipids, plates were sprayed by distilled water and white lipid spots were scraped, transferred into tubes and then lipids were extracted.

LC/ESI-MS analysis of lipids was performed on a Dionex HPLC system (utilizing the Chromeleon software), consisting of a Dionex UltiMate 3000 mobile phase pump, equipped with an UltiMate 3000 degassing unit and UltiMate 3000 autosampler (sampler chamber temperature was set at 4 °C). The Dionex HPLC system was coupled to a LXQTM ion trap mass spectrometer or to a hybrid quadrupole-orbitrap mass spectrometer, Q-Exactive (ThermoFisher, Inc., San Jose, CA) with the Xcalibur operating system. The instrument was operated in both the negative and positive ion modes (at a voltage differential of −3.5 to 5.0 kV, source temperature was maintained at 150 °C).

For phospholipid (PL) MS and MS/MS analysis of PLs was performed on a Q-Exactive hybrid-quadrupole-orbitrap mass spectrometer (ThermoFisher Inc., San Jose, CA). Analysis was performed in negative ion mode at a resolution of 140,000 for the full MS scan and 17,500 for the MS2 scan in a data-dependent mode. The scan range for MS analysis was 400–1800 m/z with a maximum injection time of 128 ms using one microscan. A maximum injection time of 500 ms was used for MS2 (high energy collisional dissociation (HCD)) analysis with collision energy set to 24. An isolation window of 1.0 Da was set for the MS and MS2 scans. Capillary spray voltage was set at 3.5 kV, and capillary temperature was 320 °C. Sheath gas was set to eight arbitrary units and the S-lens Rf level was set to 60.

**Normal phase column separation of PLs.** PLs were separated on a normal phase column (Luna 3 µm Silica (2) 100 A, 150 × 2.0 mm, (Phenomenex)) at a flow rate of 0.2 ml/min. The column was maintained at 35 °C. The analysis was performed using gradient solvents (A and B) containing 10 mM ammonium acetate and 0.5% triethylamine. Solvent A contained propanol:hexane:water (285:215:5, v/v/v) and solvent B contained propanol:hexane:water (285:215:40, v/v/v). The column was eluted for 0.5 min isocratically at 25% B, then from 0.5 to 6.5 min with a linear gradient from 25% to 40% solvent B, from 6.5 to 25 min using a linear gradient from 40 to 55% solvent B, from 25 to 38 min with a linear gradient from 55 to 70% solvent B, from 38 to 48 min using a linear gradient from 70 to 100% solvent B, then isocratically from 48 to 55 min at 100% solvent B followed by a return to initial conditions from 55 to 70 min from 100 to 25% B. The column was then equilibrated at 25% solvent B for an additional 5 min.

MS and MS/MS analysis of TAG/CE. MS and MS/MS analysis of TAG/CE was performed on a Q-Exactive hybrid-quadrupole-orbitrap mass spectrometer (ThermoFisher Inc., San Jose, CA). TAG/CE cations were formed through molecular ammonium adduction (TAG+NH4). Positional analysis of acyl chains in TAG species was performed after CID fragmentation of TAG [57, 58]. Analysis was performed in positive ion mode at a resolution of 140,000 for the full MS scan and 17,500 for the MS2 scan in a data-dependent mode with an inclusion list for TAG or CE. The scan range for MS analysis was 300–1200 m/z with a maximum injection time of 128 ms using one microscan. A maximum injection time of 500 ms was used for MS2 (high energy collisional dissociation (HCD)) analysis with collision energy set to 24. An isolation window of 1.0 Da was set for the MS and MS2 scans. Capillary spray voltage was set at 4.5 kV, and capillary temperature was 320 °C. Sheath gas was set to eight arbitrary units and the S-lens Rf level was set to 60.

Reverse phase column separation of TAG/CE. TAG/CE were separated on a reverse phase column (Luna 3 µm C18 (2) 100A, 150 × 1.0 mm, (Phenomenex)) at a flow rate of 0.065 ml/min. The column was maintained at 35 °C. The analysis was performed using gradient solvents (A and B) containing 0.1% NH4OH. Solvent A was methanol and solvent B was propanol. The column was eluted for 2 min from 0% B to 2% B (linear), from 3 to 6 min with a linear gradient from 2% solvent B to 3% solvent B, then isocratically from 3 to 18 min using 3% solvent B, 18 to 35 min with a linear gradient from 3% solvent B to 40% solvent B, 35 to 60 min using a linear gradient from 40 to 55% solvent B, then isocratically from 60 to 65 min at 55% solvent B then from 65 to 80 min from 55 to 0% B (linear) followed by equilibration from 80 to 90 min at 0% B.

**CGMD simulations of lipid droplet systems.** Coarse graining was performed using the MARTINI force field developed by Marrink et al. [59]. The lipid droplet model contained two distinct regions, phospholipid monolayer and hydrophobic core. In this model, 400 palmitoyl-oleoyl-phosphocholine (POPC) have been used to cover the core filled with 200 polyunsaturated trilinoleoyl-TAG molecules. Oxidation was introduced in the polyunsaturated-fatty-acid (PUFA) residues in 10% and in 20% of the TAG molecules. Two simulations for each system, total of six simulations, were run and analyzed to assess the number of TAG molecules that were able to leave the core region. CGMD simulations were performed using the GROMACS v. 4.5.4 MD package [60]. The system was minimized for 20 ps, before 0.2 ns NPT ensemble equilibration followed by a 0.2 ns NVT ensemble equilibration. Each MD run was carried out for 400 ns. A 20fs time step was used to integrate the equations of motion. Non-bonded interactions have a cutoff distance of 1.2 nm. Temperature and pressure were controlled using the Brendsen algorithm [61]. Simulations were run at 310 K and at 1 atm during NPT and MD runs. For all CG simulations, visualization and analysis were performed using the VMD v. 1.9 visualization software [62].

**Statistical analysis.** Statistical analysis was performed using unpaired two-tailed Student's $t$ test with significance determined at $p < 0.05$. Estimation of variation within each group of data was performed and variance was similar between groups that were compares. Animal experiments were not blinded.

**Data availability.** All relevant data are available within the articles and its supplementary informations and from the authors upon reasonable request.

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

## Acknowledgements

This work was supported by NIH grant CA165065 to D.I.G. and V.E.K. as well as flow cytometry and mouse cores of the Wistar Institute. D.I.G. was supported by Davis Family Professorship. We thank Dr. S. Hensley (University of Pennsylvania) for providing us with FLU virus and Frederick Keeney (Imaging Facility, Wistar Institute for helping with confocal microscopy analysis.

## Author contributions

Conceptualization, D.I.G. and V.E.K. Methodology, D.I.G., V.E.K., D.W., M.E.M. Validation, D.I.G. and V.E.K. Formal analysis, D.I.G., V.E.K., F.V., V.A.T. Investigation, F.V., V.A.T., D.M., M.B., E.K.D., L.D., A.K., A.A., R.A., S.P., K.A.-T., J.K.-S. Writing—original draft F.V., D.I.G., and V.E.K. Writing—review and editing, D.I.G., V.E.K., M.E.M., E.C. Funding acquisition—D.I.G. and V.E.K. Resources—M.E.M., D.W., and E.C. Supervision D.I.G. and V.E.K.

## Additional information

**Competing interests:** The authors declare no competing financial interests.

