## [Peer Review File · Nature Communications]

Reviewer #1 :
(Remarks to the Author):

The manuscript by Veglia et al. aims to characterize the mechanisms involved in the defective antigen cross-presentation by tumor associated dendritic cells. The authors provide compelling original data to support a major role for oxidized lipids in lipid droplets to bind HSP70 leading to impaired MHC trafficking and defective antigen presentation. The subject of lipid droplet functions in cancer and immunity is of great interest and the experiments presented contributes with new knowledge to the field. The manuscript is clearly written and in general well-designed and performed. However, there are specific points that need clarification to give support to the authors' conclusion. Specific points are described below:

- 1- Increased LD accumulation in DCs stimulated by tumor explant supernatants in vitro (fig 4) and in tumor bearing mice in vivo (supplemental fig 3) was demonstrated. What are the mechanisms involved in the tumor-associated increased LD formation? Does it involves increased lipogenesis and/or increased lipid uptake? Tumor bearing CD103+ DCs have defective ability to cross-present antigens and exhibit increased LDs compared to CD103- DCs. Integrins have been shown to participate in mechanisms of LD formation and metabolism (Antonov et al 2004; D'Avila et al 2011; Khalifeh-Soltani et al, 2016). Are there roles for CD103 in modulating LD in DC?
- 2- The authors used lipidomics and redox-lipidomics from total extracts of TES stimulated DCs to demonstrate differences in oxidation products in TAG and CE but not in phospholipids. As shown in fig 5, HSP70 is mainly observed in LD surface and not in LD core. Through a series of elegant predictions and assays the authors suggest that oxTAG species migrate to the LD surface where it covalently bind HSP70. Alternatively, oxidative modifications of phospholipids on LD surface could potentially occur and be overlooked. Are there differences in the pattern of phospholipid oxidation products when phospholipids from LDs and membranes were compared? It would be highly informative if lipidomics and redox lipidomics were performed in purified LDs and membrane fractions instead of total cells.
- 3- Does antioxidant treatment revert the effects of TES in triggering HSP70-LD binding and defective antigen cross-presentation in DCs?

Reviewer #2:
(Remarks to the Author):

In this manuscript, Veglia et al. define a pMHC trafficking defect in tumor-associated dendritic cells that results in impaired cross-presentation and diminished T cell stimulation. Expanding on the group's findings regarding lipid bodies in 2014, this manuscript delineates the pathway whereby oxidatively truncated lipids accumulate under tumor-bearing conditions and bind to HSP70. This binding reduces the availability of HSP70 to chaperone pMHC to the cell surface. The latter half of the manuscript contains convincing molecular biology and robustly demonstrates a role for LBs in HSP70 sequestration. However, in the first portion of the manuscript there are a number of concerning points. One of the primary weaknesses of the paper is that while the initial study presented in Figure 1 implicates CD103 DCs as the driver of priming/cross presentation defects, almost all of the subsequent studies do not use CD103 DCs. The BMDCs are not cultured in conditions that enrich of CD103 DC populations and rather rely on GM-BMDCs, which are primarily comprised of CD11b-like DCs. This results in the confounding issue that while they show data suggesting CD103- DCs are not defective in vivo, they are defective when generated in vitro. Additionally these cells are then used in vitro to study cross presentation defects with the results being generalized to represent in vivo phenomena of a different DC-type. Another focal issue is that this work is in contrast to data published by multiple other groups showing 103 DCs from tumors do in fact stimulate CD8 T cells. These authors show little to no stimulation from CD103's.

At a minimum, this inconsistency with others' published work should be addressed in the text.

This work thus seeks to expand the field's understanding of pMHC trafficking in dendritic cells and link LB accumulation and specific lipid oxidation states to tumor-derived defects in immunologic function. Beyond the points above, several major aspects of the study would benefit from clarification.

Major Issues

1. The author's need to rewrite parts of the introduction as it has apparently been copied directly from their Ramakrishnan et al., JI, 2014 paper. Major portions of the 2nd and 3rd paragraphs of the introduction are copied word-for-word and fail to reflect improved understanding that has taken place since then.
2. The switch to GM-CSF differentiated BMDCs needed to be justified by showing they are similar to the cells studied in vivo. Published work (Mayer, C.T. et. al., Blood 2014) would suggest these cells are not similar to the 103s from the mice. Additionally, the FLT3L protocol they use is not consistent with the generation of 103DCs but rather pDCs. If they have gating with additional cell surface characterization, they should include this to show 103s within the BMDC populations. It would be vastly better to use the iCD103 protocol (Mayer, C.T. et. al., Blood 2014), FL-DCs likely are not 103s making comparison to in vivo results difficult.
3. A significant proportion of GM-BMDCs are Macs (sup fig 1a) and while the text states that Macs cannot cross present because they are expressing much lower levels of pMHC (sup fig 1d.) than DCs. This data would be much stronger if the Macs were depleted/DCs were enriched in these cultures (or simply sorted out). Additionally they cite Helft et al., Immunity, 2015, which shows GM-Macs are able to cross present under these conditions.
4. Throughout the paper, they show data from in vitro BMDCs that have severe defects in cross presentation following TES culture. However, previously published work suggests that 103s from the tumor are very capable of potent T cell stim. The authors should address this in the text. It suggests that in vitro may not reflect in vivo. It is also difficult to compare the initial Fig 1 in vivo CD103 T cell stim data to the in vitro assays throughout due to different readout methods. CPM (thymidine incorporation) vs % proliferating (CFSE dilution). They should repeat the in vivo sorted DC stim assay using a consistent assay so that comparisons can be drawn.
5. The authors make the statement that pMHC accumulates and then is degraded but this is an overstatement given no data to show quantification of overall intracellular pMHC increase or evidence of degradation of the complex. Additional data should be added or overstatement removed.
6. Figure 3e, the quantification is for TES+PEC-CI which is not shown in the confocal image examples. Also, the star does not indicate where the difference lies; Ctr vs TES+PES-CI or TES vs TES+PES-CI. This would be more informative to show PES-CI image quantification on bar graph.
7. Need to show gating strategy for sort shown in figure 4e for readers to understand exactly what cells are being studied.
8. Need to sort CD103+ vs CD103- DCs from the dLNs in figure 4 rather than grouping into CD11c+ as a single population given such different phenotypes of the two cells. This would greatly strengthen the conclusions drawn from the staining.
9. For some experiments, B16F10 are used and for others LLC. While it is good to have two tumor models, they should provide data from both tumor models for each of the different cellular compartment imaging experiments rather than switching from one type to another without

justification. At a minimum, it should be made clear which is used in the text and in the figure, rather than only listing it in the legends.

10. Observed differences in HSP70 localization (Fig 4e and f) are very subtle.

11. In Fig 5, they use E. coli structure due to human being unpublished. Manuscript states that 46.5% identity and 26.3% similarity is considered "high degree of homology". If there is a citation that supports this statement for this kind of simulation trial that would be helpful reassurance. Also, if they can show that at the potential binding sites there is high sequence conservation between E. coli and human sequence, that would strengthen the argument.

Minor points:

1. Fig 1 h: it is unclear from the legend what is being shown. There is no label on the graph to distinguish long and short peptide.

2. Fig 2a: need to label colors on the histograms

3. Fig 2b: quantify the fluorescence by a meaningful metric. It is difficult to tell by eye any obvious difference though author suggests change in localization.

4. Fig 2d: unclear if graph is of one experiment or if it is three combined experiments. Needs clarification

5. Throughout the paper, figure legends should note statistical test used as well as if bars are SEM or SD.

6. Sup Fig 3c: unclear what the staining is.

7. Sup Fig 3e: DAPI needs different exposure time. Currently, it is not visible.

8. Figure 4d: The quantification graph is not explained well. Currently labeled as 'number of cells'. I is unclear if this is cells with 1 colocalization event, total colocalization, some other cut-off, etc.) Needs clarification.

9. Observed differences in HSP70 localization (Fig 4e and f) are very subtle.

10. Need quantification of figure 7a

We thank our reviewers for their thoughtful comments. We have addressed all their concerns and questions by performing a number of additional experiments. In some cases we provided explanation of the issues raised by the reviewers. We also revised the text of the manuscript to make it clearer.

Specific point-by-point responses are provided below.

Reviewer # 1

Increased LD accumulation in DCs stimulated by tumor explant supernatants in vitro (fig 4) and in tumor bearing mice in vivo (supplemental fig 3) was demonstrated. What are the mechanisms involved in the tumor-associated increased LD formation? Does it involve increased lipogenesis and/or increased lipid uptake?

We have previously demonstrated that tumor exposed DCs pick up more fluorescently labeled fatty acids than control DCs (Nat. Med. 2010, 880-886). Deletion of SRA (CD204) receptor in DCs substantially decreased lipid uptake. However, we appreciate our reviewer's point that we did not investigate lipogenesis in these cells. To address this question we studied expression of genes responsible for lipid synthesis (**Fig. R1**): acetyl-CoA-carboxylase 1 and 2 (Acc1,2, encoded by *acaca* and *acacb*), fatty acid synthase (*fasn*), Stearoyl-CoA-desaturase (*scd1*), acyl-CoA:diacylglycerol acyltransferases (*dgat1*, *dgat2*).

Editorial note: Figure R1 has been removed from this file as it contained third party images.

Figure Response 1. Schema of lipogenesis. Studied¹ members of the pathway are marked in red circle.

CD103⁺ DCs were either generated *ex vivo* in the presence of TES or isolated from lymph nodes of naïve or tumor-bearing (TB) mice. Consistent with previous observations DCs from TB mice had increased expression of *msr1* gene encoding CD204 receptor involved in lipid uptake. Expression of genes involved in lipogenesis was not increased (**Fig. R2**). Exception was *dgat2*, which was significantly up-regulated *in vitro* and *dgat1* that was up-regulated *in vivo*. However, this enzyme is involved in formation of lipid bodies directly from fatty acid that are picked up by cells. Dgat2 is found on the surface of LB (Stone SJ, et al. J Biol Chem. 2006;281:40273–40282). This supports the conclusion that lipid uptake is likely to be a major factor regulating formation of lipid bodies (LB) in DCs in TB hosts. **The results of these experiments are provided in new Supplementary Figures.**

Figure Response 2. Expression of components of lipogenesis in DCs. Upper line – CD103⁺ DCs generated *ex vivo* for 7 days using GM-CSF and FLT3-L. Lower line – CD103⁺ DCs sorted from lymph nodes of naïve and LLC TB mice (3 weeks after tumor inoculation). Indicated gene expression was evaluated by RT-qPCR and normalized to β -actin. Each group included 3 mice. *** - $p < 0.001$

Tumor bearing CD103⁺ DCs have defective ability to cross-present antigens and exhibit increased LDs compared to CD103⁻ DCs. Integrins have been shown to participate in mechanisms of LD formation and metabolism (Antonov et al 2004; D’Avila et al 2011; Khalifeh-Soltani et al, 2016). Are there roles for CD103 in modulating LD in DC?

This is a very interesting question. Although mechanism of lipid accumulation was not the focus of the study, we performed additional experiments to evaluate possible role of CD103 in LB formation and accumulation. We obtained CD103 KO mice from Jackson Lab and generated DCs from enriched HPC *in vitro* for 9 days and then exposed them to TES for 2 days as described in the manuscript. We found no differences in the total amount of lipids accumulated

in wild-type and CD103 deficient DCs. However, the proportion of DCs containing large LB ($>0.4 \mu\text{m}$) in CD103 KO DCs was smaller than in WT DCs (**Fig. R3**). These results were added to **Fig. 5**. In addition, TES did not decrease cross-presentation in CD103^{-/-} DCs in contrast to its effect on WT DCs (**Fig. R3**). These data, together with the facts that CD103⁺DCs have higher number of large LB than CD103⁻DCs and that CD103⁺ DCs have much stronger defect in cross-presentation in TB mice than CD103⁻ DCs suggest that CD103 may be involved in generation of LB in DCs and support the role of LB in defective cross-presentation in cancer.

Figure Response 3. CD103 molecule and lipid accumulation. HPC were enriched from bone marrow of wild-type (WT) and CD103^{-/-} (KO) mice. DCs were generated for 9 days with GM-CSF and FLT3-L and then exposed for 48h to TES. **a.** Phenotype of cells demonstrating lack of CD103 expression in CD11c⁺DCs. **b.** Lipid level in DCs after staining with BODIPY in DC treated with TES. **c.** CD11c⁺CD172⁻ cells were sorted and staining with BODIPY. DCs treated with TES (we show the cells treated with TES only because they have large LB) were analyzed by confocal microscopy. Typical example of staining (blue – DAPI, green – BODIPY) and the proportion of DCs with large LB ($>0.4 \mu\text{m}$) calculated per cell are shown. The number of cells counted is shown on the graph. Scale bar = $50 \mu\text{m}$. **d.** Cross-presentation of long OVA-derived peptide by DCs. * - $p < 0.05$, ***- $p < 0.001$ from WT.

2. The authors used lipidomics and redox-lipidomics from total extracts of TES stimulated DCs to demonstrate differences in oxidation products in TAG and CE but not in phospholipids. As shown in fig 5, HSP70 is mainly observed in LD surface and not in LD core. Through a series of elegant predictions and assays the authors suggest that oxTAG species migrate to the LD surface

where it covalently bind HSP70. Alternatively, oxidative modifications of phospholipids on LD surface could potentially occur and be overlooked. Are there differences in the pattern of phospholipid oxidation products when phospholipids from LDs and membranes were compared? It would be highly informative if lipidomics and redox lipidomics were performed in purified LDs and membrane fractions instead of total cells.

To address question raised by our reviewer we generated DCs *in vitro* and treated them with TES for 48 hrs. LB were isolated using gradient centrifugation and redox-lipidomics of phospholipids was performed. Characteristic of LB, PC was the major component followed by PE and several other minor classes of phospholipids (**Fig R4a**). LB PC were represented by ~30 molecular species, including several oxidizable PUFA species with linoleic (C18:2) and arachidonic (C20:4) acyls. The content of the latter (PC38:4 - 18:0/20:4), was markedly higher in TES-DC vs control DC. This PC species was also found in its oxidized mono-oxygenated form at a higher level in TES-DC vs controls (**Fig R4b**). Most importantly, neither control DC nor TES-DC contained oxidatively truncated PC or PE species readily interacting with proteins. In contrast, analysis of TAGs in LB revealed significant amounts of oxidatively truncated TAGs derivatives generated as a result of cleavage of oxygenated linoleic acid (LA) - 9-oxo-nonanoic acid (9-ONA) - and arachidonic acid (AA) - 5-hydroxy-8-oxo-6-octenoic acid (HOOA) (**Fig. R4c**). The content of these oxidatively truncated TAGs products was higher in TES-DC than in control DC and exceeded many-fold the contents of hydroxy-products in PC and PE. Based on these results, we conclude that oxidatively truncated electrophilic species capable of interacting with proteins, including HSP70, were present exclusively in LB neutral lipids. The results were added to **Supplemental Figure 8**.

3. Does antioxidant treatment revert the effects of TES in triggering HSP70-LD binding and defective antigen cross-presentation in DCs?

To address this question we used α -tocopherol (Vitamin E, Vit. E). DCs were generated with TES in the presence of Vit. E (100uM). Vit. E did not affect accumulation of lipids (**data not shown**) but decreased the number of large LB (>0.4 μ m) (**Fig. R5a**). Importantly, Vit. E abrogated defects in cross-presentation caused by TES (**Fig. R5b**). By acting as a radical scavenger, Vit E cannot affect the already formed products of lipid peroxidation. However, it can prevent the generation of peroxidized lipids, including oxidatively truncated lipids. This can explain the protective effects of Vit. E against TES induced defects in cross-presentation. **These data are included into new Fig.8.**

c

Acyl chains	m/z	Ctr DC	TES DC
16:0/18:2-OH/18:1	912.7651	—	—
16:0/9-ONA/18:1	766.6557	—	—
16:0/HOOA/18:2	766.6202	—	—
16:0/9-ONA/18:0	768.6713	—	—
18:0/9-ONA/18:1	794.6870	—	—

Figure Response 4. Redox Phospholipidomics of Lipid Bodies in DCs. DCs were generated *in vitro* from bone marrow progenitors using GM-CSF and treated with TES for 48 hrs. LB were isolated using gradient centrifugation. **a.** Distribution of phospholipid classes in lipid bodies. . PC – Phosphatidylcholine, PE – Phosphatidylethanolamine; PS - Phosphatidylserine ; PI – Phosphatidylinositol; CL – cardiolipin, PG – phosphoglycerides, BMP – bis(monoacylglycero)phosphate;, **b.** heat map of individual PC molecular species in LB from control and TES DC. **c.** Heat-map of oxidatively-truncated TAG species and hydroxy-TAG species in LD from control and TES DC.

Molecular Species	m/z	Ctr DC	TES DC
PC(30:0)	764.5439	—	—
PC(32:2)	788.5451	—	—
PC(32:1)	790.5602	—	—
PC(32:0)	792.5759	—	—
PCp(34:0)	804.6135	—	—
PC(34:2)	816.5759	—	—
PC(34:1)	818.5918	—	—
PC(34:0)	820.6092	—	—
PCp(36:3)	826.5970	—	—
PCp(36:1)	830.6266	—	—
PC(36:5)	838.5601	—	—
PC(36:4)	840.5754	—	—
PC(36:3)	842.5920	—	—
PC(36:2)	844.6066	—	—
PC(36:1)	846.6226	—	—
PC(36:0)	848.6378	—	—
PCp(38:5)	850.5957	—	—
PCp(38:4)	852.6122	—	—
PC(38:6)	864.5753	—	—
PC(38:5)	866.5914	—	—
PC(38:4)	868.6073	—	—
PC(36:3)+2O	874.5781	—	—
PC(38:4)+1O	884.6025	—	—
PC(38:3)+1O	886.6225	—	—
PC(38:2)+1O	888.6292	—	—
PC(40:6)	892.6095	—	—
PC(40:5)	894.6217	—	—
PC(38:4)+2O	900.5940	—	—

Figure Response 5. Effect of Vit. E on LD and cross-presentation by DCs. DCs were generated from HPCs with GM-CSF and FLT3-L and exposed to TES as described in manuscript. Cells were pre-incubated with Vit E and after 8h TES was added. **a.** The number of large LB per cell (CD11c+CD103+CD172a-DCs). Typical example of staining with BODIPY and the proportion of DCs with the presence of large LB (>0.4 μm) are shown. Scale bar = 50 μm. **b.** Cross-presentation of long OVA-derived peptide by DCs. *-p<0.05, ** - p<0.01, *** - p<0.001.

Reviewer #2

In this manuscript, Veglia et al. define a pMHC trafficking defect in tumor-associated dendritic cells that results in impaired cross-presentation and diminished T cell stimulation. Expanding on the group's findings regarding lipid bodies in 2014, this manuscript delineates the pathway whereby oxidatively truncated lipids accumulate under tumor-bearing conditions and bind to HSP70. This binding reduces the availability of HSP70 to chaperone pMHC to the cell surface. The latter half of the manuscript contains convincing molecular biology and robustly demonstrates a role for LBs in HSP70 sequestration. However, in the first portion of the manuscript there are a number of concerning points. One of the primary weaknesses of the paper is that while the initial study presented in Figure 1 implicates CD103 DCs as the driver of priming/cross presentation defects, almost all of the subsequent studies do not use CD103 DCs. The BMDCs are not cultured in conditions that enrich of CD103 DC populations and rather rely on GM-BMDCs, which are primarily comprised of CD11b-like DCs. This results in the confounding issue that while they show data suggesting CD103- DCs are not defective in vivo, they are defective when generated in vitro. Additionally these cells are then used in vitro to study cross presentation defects with the results being generalized to represent in vivo phenomena of a different DC-type. Another focal issue is that this work is in contrast to data published by multiple other groups showing 103 DCs from tumors do in fact stimulate CD8 T cells. These authors show little to no stimulation from CD103's. At a minimum, this inconsistency with others' published work should be addressed in the text.

The criticism of our reviewer is well taken. We addressed it directly in additional experiments as described below.

Major Issues

1. The author's need to rewrite parts of the introduction as it has apparently been copied directly from their Ramakrishnan et al., JI, 2014 paper. Major portions of the 2nd and 3rd paragraphs of the introduction are copied word-for-word and fail to reflect improved understanding that has taken place since then.

We apologize for this error. The introduction was thoroughly revised.

2. The switch to GM-CSF differentiated BMDCs needed to be justified by showing they are similar to the cells studied *in vivo*. Published work (Mayer, C.T. et. al., Blood 2014) would suggest these cells are not similar to the 103s from the mice. Additionally, the FLT3L protocol they use is not consistent with the generation of 103DCs but rather pDCs. If they have gating with additional cell surface characterization, they should include this to show 103s within the BMDC populations. It would be vastly better to use the iCD103 protocol (Mayer, C.T. et. al., Blood 2014), FL-DCs likely are not 103s making comparison to *in vivo* results difficult.

Our reviewer raised important point that we have addressed in direct experiments. However, first, it is important to point out that the issue of phenotypic characteristics of *in vitro* vs. *in vivo* DCs is far from being settled. In our study we showed that DCs generated with GM-CSF express CD135 (FLT3), which is considered as a marker associated with conventional DCs (**Fig. S1**). We used GM-CSF differentiated BMDCs for our mechanistic studies because according to a recent report (Helft et al., Immunity 2015, Immunity 2016) these DCs (CD11c⁺CD135⁺CD115⁻MHC II^{high}) have a signature of migratory DCs and could be used as model to study migratory DCs *in vitro*. According to these authors, during the culture with GM-CSF these DCs arise from committed progenitors and macrophages (CD11c⁺CD135⁻CD115⁺MHC^{int}) from monocytes. We used CD11c⁺CD135⁺CD115⁻MHC II^{high} DCs for our studies. These DCs also express CD24 and we used CD24⁺ DCs in our experiments with FLT3L culture. In these experiments, TES induced higher expression of CD103 in CD24⁺DCs (**Fig. R6**). CD24⁺DCs were sorted and then used for the experiments, in order to avoid contamination with pDC and CD11b⁺DCs. Thus, it appears that CD24⁺DCs we used to study cross-presentation are indeed bone-fide migratory DCs. In our experiments we generated DCs not from total bone marrow as in many previous studies but from enriched HPC after removal of lineage-committed cells. These cells don't contain monocytes, which made cDCs generated under these conditions even more relevant.

Figure Response 6. Expression of CD103 in CD24⁺DCs. DCs were generated from HPC by 9-days culture with FLT3L. CD24⁺ and CD11b⁺ cells were sorted and stained with indicated antibodies.

However, we agree with our reviewer suggestion and rather than discussing various data existed in the literature, we directly addressed this issue by generating CD103⁺ DCs using existing protocol (Mayer et al. Blood, 2014). We generated iCD103⁺DCs (B220⁻CD11c⁺Clec9A⁺Sirpa^{low/neg}CD103⁺Batf3⁺) with and without TES (Fig. R7a). CD103⁺ cells were sorted and used in cross-presentation experiments. CD103⁺ DCs generated with TES had defect in cross-presentation but not a direct presentation (Fig. R7b). In CD103⁺ DCs treated with TES, but not in control DCs, pMHC was co-localized with lysosomes (Fig. R7c). LB accumulated in TES treated DCs co-localized with HSP70 (Fig. R7d). Thus, these results indicate that the TES effects on cross-presenting ability of DCs are reproduced in in CD103⁺ DCs generated *in vitro*. **These data are included to Fig. 1, 4, and Supplementary Fig. 3**

Figure Response 7. Effect of TES on CD103⁺ DCs. DCs were generated *in vitro* from bone marrow progenitors using combination of GM-CSF and FLT3L. Cells were treated with TES for 48h. **a.** Proportion of CD103⁺ DCs generated. **b.** CD103⁺ DCs were sorted and used in cross-presentation of long peptide or direct presentation of short peptide. Cumulative results of three experiments are shown. **c,d.** Confocal microscopy of sorted CD103⁺ DCs loaded with long peptide. Scale bars = 50 μ m

3. A significant proportion of GM-BMDCs are Macs (sup fig 1a) and while the text states that Macs cannot cross present because they are expressing much lower levels of pMHC (sup fig 1d.) than DCs. This data would be much stronger if the Macs were depleted/DCs were enriched in these cultures (or simply sorted out). Additionally they cite Helft et al., Immunity, 2015, which shows GM-Macs are able to cross present under these conditions.

We have performed additional experiments to address this question. We generated DCs and Macrophages (Mac) by using GM-CSF (Helft et al., Immunity, 2015) (**Fig. R8a**) and showed that TES caused substantial decrease in pMHC expression on the surface of both DCs and Mac after loading cells with long peptide (**Fig. R8b**) Similar effect was observed in stimulation of specific CD8⁺ T cell proliferation (**Fig. R8c**). It is important to point out that both expression of pMHC and stimulation of CD8⁺ T cells after cross-presentation by DCs was almost 10-fold higher than by Mac, which supports overall critical role of DCs in cross-presentation. No effect of TES on direct presentation by DCs or Mac was seen (**Fig. R8d**). **The results of these experiments were included Supplemental Figure 2.**

Figure Response 8. Effect of TES on cross-presentation by macrophages and DCs. DCs and macrophages (Mac) were generated from bone marrow HPC for 5 days with GM-CSF and then treated with TES for 48 hours. Cells were loaded with OVA derived long or short peptides and used in the experiments. **a.** Gating of DCs and Mac. **b.** Expression of pMHC on the surface of DCs and Mac. Typical example of three performed experiments is shown. **c.** Proliferation of OT-1 CD8⁺ transgenic T cells stimulated with DCs or Mac loaded with OVA-derived long peptide. Proliferation was measured in triplicates. Three experiments were performed. * - p<0.05; ** - p<0.01 from control. **c. d.** Proliferation of OT-1 CD8⁺ transgenic T cells stimulated with DCs or Mac loaded with OVA-derived short peptide. Three experiments were performed.

4. Throughout the paper, they show data from *in vitro* BMDCs that have severe defects in cross presentation following TES culture. However, previously published work suggests that 103s from the tumor are very capable of potent T cell stim. The authors should address this in the text. It suggests that *in vitro* may not reflect *in vivo*. It is also difficult to compare the initial Fig 1 *in vivo* CD103 T cell stim data to the *in vitro* assays throughout due to different readout methods. CPM (thymidine incorporation) vs % proliferating (CFSE dilution). They should repeat the *in vivo* sorted DC stim assay using a consistent assay so that comparisons can be drawn.

Previously published works suggested that CD103⁺ DCs can cross-present antigens in cancer. However, the efficacy of such cross-presentation was not ascertained since in those studies control DCs were not evaluated in the same experimental system. We argue that the fact that in tumor-bearing hosts immune response is very difficult to generate may suggest that cross-presentation by DCs may not be efficient. Many mechanisms may contribute to immune defects in cancer. Therefore, in this study, we directly compared cross-presentation by DCs between control (tumor free) and tumor-bearing mice under similar experimental conditions *in vivo* and demonstrated that DCs in tumor-bearing mice had significantly lower ability to cross-present antigens than their counterparts in tumor-free mice. We provided more detailed discussion of this point in the paper. Direct comparison between values obtained *in vivo* and *in vitro* experiments is not informative since the conditions are vastly different between DCs directly obtained from tissues and generated from HPC. Experiments *in vivo* were performed with CD103⁺ DCs sorted directly from tissues. Biological replicates were represented by individual mice. The number of cells available from each mouse was limited; therefore standard thymidine incorporation assay was technically difficult to perform. I hope our reviewer agree that under these conditions we select best method that allowed for detection of T-cell response.

5. The authors make the statement that pMHC accumulates and then is degraded but this is an overstatement given no data to show quantification of overall intracellular pMHC increase or evidence of degradation of the complex. Additional data should be added or overstatement removed.

We agree with this comment and removed this overstatement.

6. Figure 3e, the quantification is for TES+PEC-Cl which is not shown in the confocal image examples. Also, the star does not indicated where the difference lies; Ctr vs TES+PES-Cl or TES vs TES+PES-Cl. This would be more informative to show PES-Cl image quantification on bar graph.

We apologize for the mistake with labeling on the graph. PES-Cl was used to mimic the effect of TES (by blocking HSP70 activity) and thus was never combined with TES since it was not scientifically justified. * indicate differences from control for both, TES and PES-Cl treated DCs.

We have revised this figure.

7. Need to show gating strategy for sort shown in figure 4e for readers to understand exactly what cells are being studied.

In Fig. 4e and f we used DC generated with GM-CSF. Sorted strategy is shown in **Supplemental Figure 1**.

8. Need to sort $CD103^+$ vs $CD103^-$ DCs from the dLNs in figure 4 rather than grouping into $CD11c^+$ as a single population given such different phenotypes of the two cells. This would greatly strengthen the conclusions drawn from the staining.

This point is well taken. We perform additional experiments where we sorted $CD103^+$ and $CD103^-$ DCs and evaluated co-localization between lipid droplets with HSP70. These results are added to **Fig. 4** in new version of manuscript and provided below in **Fig. R9**

Figure Response 9. Co-localization of lipid bodies and HSP70 in DCs. $CD103^+$ and $CD103^-$ DCs were sorted from dLNs of tumor bearing mice. On the left –typical example of staining. On the right – proportion of co-localization of lipid bodies with HSP70. Four independent experiments were performed. *** - $p < 0.001$.

9. For some experiments, B16F10 are used and for others LLC. While it is good to have two tumor models, they should provide data from both tumor models for each of the different cellular compartment imaging experiments rather than switching from one type to another without justification. At a minimum, it should be made clear which is used in the text and in the figure, rather than only listing it in the legends.

We have revised text to provide clear information about the tumor models used. We used EL4 and LLC. For *ex vivo* experiment we used LLC because EL4 metastasizes to lymph nodes, which makes evaluation of DC difficult.

10. Observed differences in HSP70 localization (Fig 4e and f) are very subtle.

We appreciate this concern. However, in biological system with active cell function, it is not surprising that differences are not dramatic. After all, we observed only 2-3 fold decrease in cross-presentation, which is much more sensitive method for detection. However, it may have profound biological impact.

11. In Fig 5, they use E. coli structure due to human being unpublished. Manuscript states that 46.5% identity and 26.3% similarity is considered "high degree of homology". If there is a citation that supports this statement for this kind of simulation trial that would be helpful reassurance. Also, if they can show that at the potential binding sites there is high sequence conservation between E. coli and human sequence, that would strengthen the argument.

We revised manuscript to provide more information. A successful approach in modeling the organization of a previously unannotated protein is via employment of the sequence similarity with a protein with known structure and functions. Since the crystal structure of the human HSP70 has not been published, we utilized the *E. coli* HSP70, which has a high degree of similarity to human HSP70. The principle "Sequence alignments unambiguously distinguish between protein pairs of similar and non-similar structure when the pairwise sequence identity is high (>40% for long alignments)" is commonly accepted in the field Rost B, Twilight zone of protein sequence alignments, Protein Eng Des Sel (1999) 12 (2): 85-94. In our case, the two proteins fulfill the requirement as they have 46.5% identity and 26.3% similarity representing a high degree of homology, It should be also noted that we employed BLAST (Basic Local Alignment Search Tool) to align the two sequences. The Expect value (E-value) characterizes the significance of the sequence match. The lower the E-value, or the closer it is to zero, the more "significant" the match is. In the case of *E. coli* HSP70 and human HSP70, the E-value is equal to 0.0 demonstrating a high level of sequence similarity.

To address the concern related to sequence conservation between the *E. coli* and human HSP70 proteins, we aligned these two sequences to investigate Sites 1, 2, and 3. As shown below, Sites 1 and 2 contain two conserved lysine residues (shown in red) that are predicted to be responsible for the covalent immobilization of HSP70 on the ox-TAG-LB surface. These lysine residues can chemically interact with the reactive electrophilic group of oxidatively truncated lipids. Therefore, HSP70 can be anchored on the surface of ox-TAG-LB through a two-step process: i) the initial interactions governed by long-range non-bonded forces including electrostatic and hydrophobic forces, ii) short-range chemical interactions leading to lipidation of HSP70. With regards to site 3, it does not include any conserved lysine residues. However, it comprises 10 hydrophobic amino acid residues (45%, shown in red). We assigned 15Å penetration of this site into the ox-TAG-LB based on the high level of hydrophobicity of this site likely leading to non-covalent stabilization of HSP70 on the surface of ox-TAG-LB.

Site 1:

E coli: (30) AEG----- (43)TQDGE - (49)L - - (52)QPA**K**RQAVTN

Human: (33) DQG----- (46)T- DTE - (52)L - - (55)DAA**K**NQVALN

Site 2:

E coli: (244)**F**KKDQGI DLRNDPLAMQRLKE

Human: (249)**F**KRKHKKDI SQNKRAV RRLRT

Site 3:

E coli: (279) TD**V**NLPYITAD**A**TGPKHM**N**IK**V**

Human: (284) AS LE IDSLFEG I - - - - DF YTSI

Minor points:

1. *Fig 1 h: it is unclear from the legend what is being shown. There is no label on the graph to distinguish long and short peptide.*

We have revised this panel to make it clear. Cross-presentation indicates the use of long peptide and direct binding indicate use of short peptide.

2. *Fig 2a: need to label colors on the histograms*

Colors on histogram are now labeled.

3. *Fig 2b: quantify the fluorescence by a meaningful metric. It is difficult to tell by eye any obvious difference though author suggests change in localization.*

Quantification in this case is very difficult to do since we demonstrate membrane localization of the staining in control DCs and cytoplasmic localization in TES treated DCs. We hope that differences are visible. However, we understand the issues and increased the size and the quality of staining and expanded description of the experiment in the text.

4. *Fig 2d: unclear if graph is of one experiment or if it is three combined experiments. Needs clarification*

It is a representative of three experiments performed. Clarification is provided.

5. *Throughout the paper, figure legends should note statistical test used as well as if bars are SEM or SD.*

We corrected this error. We provided information about the use of SD in figure legend. We also made sure that we clarify statistical method used in all figures.

6. *Sup Fig 3c: unclear what the staining is.*

This staining was for lipids using BODIPY. We have provided clarification.

7. *Sup Fig 3e: DAPI needs different exposure time. Currently, it is not visible.*

We have revised that figure. We increased the quality and dapi is much more visible.

8. *Figure 4d: The quantification graph is not explained well. Currently labeled as 'number of cells'. I is unclear if this is cells with 1 colocalization event, total colocalization, some other cut-off, etc.) Needs clarification.*

We have provided clarification. We counted the cells showing co-localization HSP70-LB out of 100 cells counted.

9. Observed differences in HSP70 localization (Fig 4e and f) are very subtle.

We appreciate this concern. However, in biological system with active cell function, it is not surprising that differences are not dramatic. After all, we observed only 2-3 fold decrease in cross-presentation, which is much more sensitive method for detection. However, it may have profound biological impact.

Reviewer #1 (Remarks to the Author):

The authors have adequately addressed the concerns and questions raised.

Reviewer #2 (Remarks to the Author):

The authors have addressed all of our queries.